# Gradient Testing and Estimation by Comparisons

**Xiwen Tao** [* 1 2]  **Chenyi Zhang** [* 3]  **Helin Wang** [1 2]  **Yexin Zhang** [2 4]  **Tongyang Li** [2 4]

## Abstract

We study gradient testing and gradient estimation of smooth functions using only a comparison oracle that, given two points, indicates which one has the larger function value. For any smooth $f\colon \mathbb{R}^n \to \mathbb{R}$, $\mathbf{x} \in \mathbb{R}^n$, and $\varepsilon > 0$, we design a gradient testing algorithm that determines whether the normalized gradient $\nabla f(\mathbf{x})/\|\nabla f(\mathbf{x})\|$ is $\varepsilon$-close or $2\varepsilon$-far from a given unit vector $\mathbf{v}$ using $O(1)$ queries, as well as a gradient estimation algorithm that outputs an $\varepsilon$-estimate of $\nabla f(\mathbf{x})/\|\nabla f(\mathbf{x})\|$ using $O(n \log(1/\varepsilon))$ queries which we prove to be optimal. Furthermore, we study gradient estimation in the quantum comparison oracle model where queries can be made in superpositions, and develop a quantum algorithm using $O(\log(n/\varepsilon))$ queries.

## 1. Introduction

Optimization is foundational to machine learning, since the training of a neural network is equivalent to the minimization of its loss function. However, there are common scenarios where access to gradients is infeasible or computationally prohibitive, such as black-box adversarial attack on neural networks (Papernot et al., 2017; Madry et al., 2018; Chen et al., 2017) and policy search in reinforcement learning (Salimans et al., 2017; Choromanski et al., 2018). This motivates the study of optimization algorithms that operate with zeroth-order information, i.e., function-value evaluations. A series of work has established rigorous convergence and complexity guarantees for such methods in both convex optimization (Duchi et al., 2015; Nesterov & Spokoiny, 2017) and nonconvex optimization (Ghadimi & Lan, 2013; Fang et al., 2018; Jin et al., 2018; Ji et al., 2019; Zhang et al.,

2022; Vlatakis-Gkaragkounis et al., 2019; Balasubramanian & Ghadimi, 2022).

On the other hand, following the rapid advancement of quantum computing (Preskill, 2018; 2025), quantum algorithms that provide speedup for solving optimization problems have been systematically investigated. These include constrained optimization problems covering linear programs (Casares & Martin-Delgado, 2020; Bouland et al., 2023; Gao et al., 2023), second-order cone programming (Kerenidis et al., 2021; Garrido et al., 2025), and semidefinite programs (Brandão & Svore, 2017; Brandão et al., 2019; van Apeldoorn & Gilyén, 2019; van Apeldoorn et al., 2020b). General convex optimization (Chakrabarti et al., 2020; van Apeldoorn et al., 2020a; Sidford & Zhang, 2023) and nonconvex optimization (Zhang et al., 2021; Liu et al., 2023; Leng et al., 2023; Chen et al., 2025; Leng et al., 2025) are also extensively studied. See the survey by Dalzell et al. (2023) for more details.

More recently, optimization algorithms are soliciting for even less information. In particular, an intriguing setting is to only have access to comparisons of function values. Formally, for a function $f\colon \mathbb{R}^n \to \mathbb{R}$, its comparison oracle is defined as $O_f^{\mathrm{comp}}\colon \mathbb{R}^n \times \mathbb{R}^n \to \{-1, 1\}$ that for any pair of inputs $(x, y) \in \mathbb{R}^n \times \mathbb{R}^n$, it satisfies

$$O_f^{\mathrm{comp}} = \begin{cases} 1 & f(x) \geq f(y) \\ -1 & f(x) \leq f(y) \end{cases}. \tag{1}$$

(When $f(\mathbf{x}) = f(\mathbf{y})$, it is allowed to output either 1 or $-1$.)

Comparisons have long been used as a basic primitive in derivative-free optimization; see the survey by Larson et al. 2019. Classical direct-search and pattern-search methods compare candidate points to decide whether to accept a trial step (Kolda et al., 2003; Audet & Dennis Jr, 2006), and the Nelder–Mead method (Nelder & Mead, 1965) iteratively compares candidate points to seek lower objective values. However, the Nelder–Mead method may fail to converge to a stationary point even for smooth functions (McKinnon, 1998). More recent works study comparison-based update rules with provable guarantees. The Stochastic Three-Point method (Bergou et al., 2020) and related stochastic derivative-free methods (Gorbunov et al., 2020) sample random search directions and use comparisons among trial points to select the next iterate. GradientLess De-

---

[*]Equal contribution  [1]School of Electronics Engineering and Computer Science, Peking University [2]Center on Frontiers of Computing Studies, Peking University [3]Computer Science Department, Stanford University [4]School of Computer Science, Peking University. Correspondence to: Tongyang Li <tongyangli@pku.edu.cn>.

*Proceedings of the $43^{rd}$ International Conference on Machine Learning*, Seoul, South Korea. PMLR 306, 2026. Copyright 2026 by the author(s).

scent (Golovin et al., 2020) develops comparison-invariant zeroth-order algorithms with improved dimension dependence under structural assumptions. Jamieson et al. (2012) proved lower bounds for noisy derivative-free optimization and showed that, for smooth strongly convex objectives, Boolean comparison feedback can match the convergence rate of noisy function-value feedback. Tang et al. (2024) studied ranking-based variants of the comparison model, where the oracle returns an ordering over multiple candidate points.

Despite the importance of gradients in optimization, prior work has not pinned down the optimal algorithm for gradient estimation using comparison queries. An existing result is due to Karabag et al. (2021), which achieved $O(n^2 \log n)$ comparison queries per gradient direction estimate at a fixed angular accuracy. Another related work is dueling optimization (Saha et al., 2021), which provided convergence guarantees based on update directions that are aligned with the true gradient in expectation; however, their analysis does not characterize the query complexity required to estimate gradient directions to a designated accuracy. In a different line of work, Cai et al. (2022) leveraged ideas from one-bit compressed sensing to recover gradient information from noisy comparison feedback.

Beyond that, gradient testing is also a natural problem to explore. In general, property testing decides whether a given object has a certain property or is significantly different from any object that has the property. It typically brings more efficient algorithms than property estimation, and may circumvent significant costs when the input size is large or an exact algorithm is expensive. Therefore, property testing is a main algorithmic topic (Ron, 2010; Goldreich, 2017), especially on discrete problems. However, its study on continuous optimization is scarce, which motivates us to study gradient testing together with gradient estimation.

In the quantum setting, it is also very natural to study optimization by comparisons, particularly because current quantum computers are noisy and simpler information directly benefits implementation of quantum algorithms in the near term. However, quantum optimization algorithms by comparisons with quantum advantages remain widely open.

**Results.** In this paper, we systematically study gradient testing and estimation, formally stated as follows:

**Problem 1.1** (Gradient testing). Let $f\colon \mathbb{R}^n \to \mathbb{R}$ be $L$-smooth, $\varepsilon \in (0, 1/\sqrt{2})$ and $\gamma > 0$ be fixed parameters. Given query access to its comparison oracle $\mathcal{O}_f^{\mathrm{comp}}$ (Eq. (1)), a point $\mathbf{x} \in \mathbb{R}^n$, and a unit vector $\mathbf{v} \in \mathbb{R}^n$, the goal is to decide whether

$$\left\| \frac{\nabla f(\mathbf{x})}{\|\nabla f(\mathbf{x})\|} - \mathbf{v} \right\| \leq \varepsilon \quad \text{or} \quad \left\| \frac{\nabla f(\mathbf{x})}{\|\nabla f(\mathbf{x})\|} - \mathbf{v} \right\| > 2\varepsilon,$$

promised that one of the two cases holds and $\|\nabla f(\mathbf{x})\| \geq \gamma$.

**Problem 1.2** (Gradient estimation). Let $f\colon \mathbb{R}^n \to \mathbb{R}$ be $L$-smooth, $\varepsilon \in (0, 1/\sqrt{2})$ and $\gamma > 0$ be fixed parameters. Given query access to its comparison oracle $\mathcal{O}_f^{\mathrm{comp}}$ (Eq. (1)) and a point $\mathbf{x} \in \mathbb{R}^n$, the goal is to output a unit vector $\mathbf{v} \in \mathbb{R}^n$ satisfying $\left\| \frac{\nabla f(\mathbf{x})}{\|\nabla f(\mathbf{x})\|} - \mathbf{v} \right\| \leq \varepsilon$, promised that $\|\nabla f(\mathbf{x})\| \geq \gamma$.

Both problems are scale-invariant: multiplying the objective by a positive constant does not change either the comparison oracle or the normalized gradient direction. The lower bound $\|\nabla f(\mathbf{x})\| \geq \gamma$ is therefore a nondegeneracy condition rather than an additional source of query complexity. It is necessary because the normalized direction is undefined at stationary points, and, when the gradient norm is arbitrarily small, the first-order term used by directional-preference comparisons can be dominated by the $L$-smoothness error. In our algorithms, $\gamma$ only determines the probing radius used in each comparison; the number of comparison queries is independent of the numerical value of $\gamma$. Once an estimate $\mathbf{h} \approx \nabla f(\mathbf{x})/\|\nabla f(\mathbf{x})\|$ is obtained, it can be used directly in normalized gradient descent updates of the form $\mathbf{x}_{t+1} = \mathbf{x}_t - \eta_t \mathbf{h}_t$, so the local estimation primitive can be plugged into comparison-based optimization procedures. Specific optimized experiment can be seen in Section 5.

For these two problems, we obtained classical and quantum algorithms with comparison query complexities as follows:

- Classical gradient testing (Algorithm 2): $O(1)$

- Classical gradient estimation (Algorithm 3): $O(n \log \frac{1}{\varepsilon})$

- Quantum gradient estimation (Algorithm 4): $O(\log \frac{n}{\varepsilon})$

In addition, we prove that *our algorithms for the first three problems are optimal*. For quantum gradient estimation, we prove that $\Omega(\log(\frac{1}{\varepsilon}))$ is a quantum lower bound, which is optimal up to a factor of $\log n$. See a summary in Table 1.

## 2. Preliminaries

### 2.1. Notations

Let $f\colon \mathbb{R}^n \to \mathbb{R}$ be a differentiable $L$-smooth function, i.e.,

$$\|\nabla f(\mathbf{x}) - \nabla f(\mathbf{y})\| \leq L\|\mathbf{x} - \mathbf{y}\| \quad \forall \mathbf{x}, \mathbf{y} \in \mathbb{R}^n.$$

For any vector $\mathbf{v}$, $\|\mathbf{v}\|$ represents its $\ell_2$-norm. For any $\mathbf{x} \in \mathbb{R}^n$, $\mathbf{g} = \nabla f(\mathbf{x})/\|\nabla f(\mathbf{x})\|$ represents its gradient direction. We denote $\mathbf{e}_i$ to be the unit vector of the $i$-th coordinate component, and use $g_i$ to represent the value of $\mathbf{g}$ at the $i$-th coordinate component. We denote $S_n := \{\mathbf{x} \in \mathbb{R}^n \colon \|\mathbf{x}\| = 1\}$ to be the unit sphere in $\mathbb{R}^n$, and $\mathbf{x} \sim S_n$ means that $\mathbf{x}$ is drawn from the uniform distribution on $S_n$.

*Table 1.* Summary of our main results: classical and quantum algorithms for gradient testing and estimation by comparisons.

|  | Gradient Testing | Gradient Estimation |
|---|---|---|
| Classical | $\Theta(1)$ (Theorem 3.2) | $\Theta(n \log \frac{1}{\varepsilon})$ (Theorem 3.4 and Theorem 6.1) |
| Quantum |  | $O(\log \frac{n}{\varepsilon})$ (Theorem 4.1), $\Omega(\log \frac{1}{\varepsilon})$ (Theorem 6.2) |

## 2.2. Inner Product Concentration for Random Vectors

Throughout the paper, we use the following estimation of the inner product between a fixed unit vector and a unit vector selected from $S_n$ uniformly at random:

**Lemma 2.1.** *Let $n \geq 5$. For any $\mathbf{x} \in \mathbb{R}^n$, $\mathbf{x} \neq \mathbf{0}$, and any constant $c > 0$, there exist constants $p_1$ and $p_2$ such that*

$$p_1 \leq \Pr_{\mathbf{y} \sim S_n}\left[|\langle \mathbf{y}, \mathbf{x} \rangle| \leq \|\mathbf{x}\|/(c\sqrt{n})\right] \leq p_2,$$

*where $\mathbf{y}$ is chosen from $S_n$ uniformly at random. In particular, we have the inequalities below:*

- $\Pr_{\mathbf{y} \sim S_n}[|\langle \mathbf{y}, \mathbf{x} \rangle| \leq \|\mathbf{x}\| \cdot \frac{24}{25\sqrt{n}}] \geq 3/5$

- $\Pr_{\mathbf{y} \sim S_n}[|\langle \mathbf{y}, \mathbf{x} \rangle| \leq \|\mathbf{x}\| \cdot \frac{18}{25\sqrt{n}}] \leq 11/20$

- $\Pr_{\mathbf{y} \sim S_n}[|\langle \mathbf{y}, \mathbf{x} \rangle| \geq \|\mathbf{x}\|/(5\sqrt{n})] \geq 4/5$

The proof of Lemma 2.1 is deferred to Appendix A.1.

## 2.3. Basic Definitions in Quantum Computing

We briefly introduce basic notions from quantum computing that are relevant to this paper; a comprehensive introduction can be found from the textbook by Nielsen & Chuang (2010). We use Dirac notation $|\cdot\rangle$ to represent quantum states, which can be seen as column vectors. The basic unit of quantum state, qubit, is represented by a normalized vector in $\mathbb{C}^2$, which has the superposition form $|\phi\rangle = \alpha|0\rangle + \beta|1\rangle$ on the computational basis $\{|0\rangle, |1\rangle\}$ with $\alpha, \beta \in \mathbb{C}$ and $|\alpha|^2 + |\beta|^2 = 1$. An $n$-qubit system is represented in a $2^n$-dimensional complex vector space, with basis states $\{|0\rangle, |1\rangle\}^{\otimes n}$. A general $n$-qubit state can be expressed as $|\psi\rangle = \sum_{x \in \{0,1\}^n} \alpha_x |x\rangle$ with $\sum_x |\alpha_x|^2 = 1$. Quantum measurement on the computational basis collapses the state $|\psi\rangle = \sum_x \alpha_x |x\rangle$ to one of the basis states $|x\rangle$ with probability $|\alpha_x|^2$. Quantum operations are modeled as unitary transformations, which are linear operators $U$ satisfying $U^\dagger U = I$, where $U^\dagger$ is the conjugate transpose of $U$.

A quantum algorithm can access a function via queries to a quantum oracle. For a classical function $g$, the oracle $O_g$ is a unitary transformation that maps $|x\rangle|b\rangle$ to $|x\rangle|b \oplus g(x)\rangle$. This allows the oracle to be queried on superpositions of inputs, producing corresponding superpositions of outputs.

**Quantum Fourier transform (QFT)** is the quantum analogue of the discrete Fourier transform, but operates on amplitudes of quantum states. Formally, given the computational basis $\{|0\rangle, |1\rangle, \ldots, |N-1\rangle\}$, it acts as follows:

$$\text{QFT}: |x\rangle \mapsto \frac{1}{\sqrt{N}} \sum_{k=0}^{N-1} e^{2\pi i x k/N} |k\rangle.$$

A key application of QFT is Jordan's gradient estimation algorithm (Jordan, 2005). Specifically, if the phase of each basis state is given by an inner product, then applying the inverse QFT and measuring in the computational basis allows one to recover the underlying vector with high probability.

**Proposition 2.2** (informal version of Proposition C.1). *Let $n \geq 2$ and $t \in \mathbb{Z}_+$. There exists a quantum procedure that, given an input state encoding $\mathbf{x} \in [0,1]^n$ as $\frac{1}{\sqrt{(t+1)^n}} \sum_{\mathbf{y} \in \{0,1,\ldots,t\}^n} e^{2\pi i \langle \mathbf{y}, \mathbf{x} \rangle} |\mathbf{y}\rangle$, outputs an estimate $\hat{\mathbf{v}} \in \{0, 1/t, \ldots, 1\}^n$ such that*

$$\Pr\left[\|\hat{\mathbf{v}} - \mathbf{x}\|_2 \leq O(n^{1.5}/t)\right] \geq 2/3.$$

*If the input state only approximates the ideal encoding with error $\varepsilon$, the success probability decreases by at most $O(\varepsilon)$.*

A formal statement with proof is deferred to Appendix C.

## 3. Classical Algorithms

### 3.1. Directional Preference

First, we show that given a point $\mathbf{x} \in \mathbb{R}^n$ and a direction $\mathbf{v} \in \mathbb{R}^n$, we can use one comparison query to understand whether the inner product $\langle \nabla f(\mathbf{x}), \mathbf{v} \rangle$ is roughly positive or negative, which determines whether $\mathbf{v}$ is following or against the direction of $\nabla f(\mathbf{x})$. This is also known as *directional preference* (DP) in Karabag et al. (2021).

**Lemma 3.1.** *Let $f: \mathbb{R}^d \to \mathbb{R}$ be an $L$-smooth function. Given a point $\mathbf{x} \in \mathbb{R}^d$, a unit vector $\mathbf{v} \in B_1(0)$, and precision $\Delta > 0$ for directional preference. Then we have:*

- *If $O_f^{\text{comp}}(\mathbf{x} + \frac{2\Delta}{L}\mathbf{v}, \mathbf{x}) = 1$, then $\langle \nabla f(\mathbf{x}), \mathbf{v} \rangle \geq -\Delta$.*

- *If $O_f^{\text{comp}}(\mathbf{x} + \frac{2\Delta}{L}\mathbf{v}, \mathbf{x}) = -1$, then $\langle \nabla f(\mathbf{x}), \mathbf{v} \rangle \leq \Delta$.*

*Proof.* Since $f$ is an $L$-smooth differentiable function,

$$\left|f(\mathbf{y}) - f(\mathbf{x}) - \langle \nabla f(\mathbf{x}), \mathbf{y} - \mathbf{x} \rangle\right| \leq \frac{1}{2}L\|\mathbf{y} - \mathbf{x}\|^2$$

**Algorithm 1** DP($\mathbf{x}, \mathbf{v}, \Delta$)

1: **Input:** Comparison oracle $O_f^{\text{comp}}$ for $f\colon \mathbb{R}^n \to \mathbb{R}$, point $\mathbf{x} \in \mathbb{R}^n$, unit vector $\mathbf{v} \in B_1(0)$, parameter $\Delta > 0$
2: **if** $O_f^{\text{comp}}\left(\mathbf{x} + \frac{2\Delta}{L}\mathbf{v},\ \mathbf{x}\right) = 1$ **then**
3:    **Return:** "$\langle \nabla f(\mathbf{x}), \mathbf{v} \rangle \geq -\Delta$"
4: **else**
5:    **Return:** "$\langle \nabla f(\mathbf{x}), \mathbf{v} \rangle \leq \Delta$"

for any $\mathbf{x}, \mathbf{y} \in \mathbb{R}^n$. Take $\mathbf{y} = \mathbf{x} + \frac{2\Delta}{L}\mathbf{v}$, this gives

$$f(\mathbf{y}) - f(\mathbf{x}) - \frac{2\Delta}{L}\langle \nabla f(\mathbf{x}), \mathbf{v} \rangle \leq \frac{1}{2}L\left(\frac{2\Delta}{L}\right)^2 = \frac{2\Delta^2}{L}.$$

Therefore, if $O_f^{\text{comp}}(\mathbf{y}, \mathbf{x}) = 1$, i.e., $f(\mathbf{y}) \geq f(\mathbf{x})$,

$$\frac{2\Delta}{L}\langle \nabla f(\mathbf{x}), v \rangle \geq \frac{2\Delta}{L}\langle \nabla f(\mathbf{x}), \mathbf{v} \rangle + f(\mathbf{x}) - f(\mathbf{y}) \geq -\frac{2\Delta^2}{L},$$

and hence

$$\langle \nabla f(\mathbf{x}), \mathbf{v} \rangle \geq -\Delta.$$

On the other hand, if $O_f^{\text{comp}}(\mathbf{y}, \mathbf{x}) = -1$, i.e., $f(\mathbf{y}) \leq f(\mathbf{x})$, we have

$$\frac{2\Delta}{L}\langle \nabla f(\mathbf{x}), \mathbf{v} \rangle \leq f(\mathbf{y}) - f(\mathbf{x}) + \frac{2\Delta^2}{L} \leq \frac{2\Delta^2}{L},$$

and hence $\langle \nabla f(\mathbf{x}), \mathbf{v} \rangle \leq \Delta$. $\qquad \square$

### 3.2. Gradient Testing

Here we present our algorithm for gradient testing (Problem 1.1). Without loss of generality, we assume the given unit vector $\mathbf{v}$ satisfies $\mathbf{v} = (1, 0, \ldots, 0)^\top$.

**Algorithm 2** Classical Gradient Testing

1: **Input:** testing direction $\mathbf{v}$, lower bound $\gamma$ on $\|\nabla f\|$
2: Set $\Delta = \varepsilon/(25\sqrt{2}n)$, $\delta = 1/3$, $T = \lceil 3200 \ln(1/\delta) \rceil$
    Initialize $N = 0$
3: **for** $i = 1$ *to* $T$ **do**
4:    Randomly choose $\mathbf{y} \sim S_{n-1}$ and define $\alpha_{\mathbf{y}} = (-\varepsilon/\sqrt{(n-1)(1-\varepsilon^2)}, \mathbf{y})$
5:    Call Algorithm 1 with input $(\mathbf{x}, \alpha_{\mathbf{y}}/\|\alpha_{\mathbf{y}}\|, \Delta)$
6:    **if** $\langle \nabla f, \alpha_{\mathbf{y}}/\|\alpha_{\mathbf{y}}\| \rangle \leq \Delta$ **then**
7:       $N \leftarrow N + 1$
8: **if** $N/T \geq 63/80$ **then**
9:    **Return:** Yes
10: **else**
11:    **Return:** No

The high-level idea of Algorithm 2 is as follows. For our convenience, we decompose $\mathbf{g}$ as $\mathbf{g} = (g_1, \tilde{\mathbf{g}})$ and denote $\mathbf{u} = \mathbf{v} - \mathbf{g}$. Note that $\langle \tilde{\mathbf{g}}, \mathbf{y} \rangle$ is related to the length of the norm $\|\tilde{\mathbf{g}}\|$. If we set a comparison standard $k$, then the

probability of $\mathbf{y}$ such that the inner product is less than $k$ is monotonically increasing with respect to $\|\tilde{\mathbf{g}}\|$. Specifically, we set $k = \varepsilon/\sqrt{n-1}$ here and in this case, there is a constant difference between the different probability. As a result, we can randomly choose a constant number of $\mathbf{y}$ to distinguish them.

**Theorem 3.2.** *Algorithm 2 solves Problem 1.1 using $O(1)$ queries to a comparison oracle $O_f^{\text{Comp}}$ in Eq. (1) with success probability at least $\frac{2}{3}$.*

*Proof of Theorem 3.2.* Since $\|\mathbf{g}\| = 1$, we have

$$\|\mathbf{g} - \mathbf{e}_1\|^2 = (1 - g_1)^2 + g_2^2 + \cdots + g_n^2$$
$$= (1 - g_1)^2 + 1 - g_1^2 = 2 - 2g_1.$$

If $\|\mathbf{g} - \mathbf{e}_1\| \leq \varepsilon$, we can get $g_1 \geq 1 - \varepsilon^2/2$, which means

$$\|\tilde{\mathbf{g}}\|^2 = g_1^2 \sum_{i=2}^n \alpha_i^2 = 1 - g_1^2$$
$$\leq 1 - (1 - \varepsilon^2/2)^2 = \varepsilon^2 - \varepsilon^4/4 \leq \varepsilon^2, \quad (2)$$

and

$$\varepsilon\langle \mathbf{v}, \mathbf{g} \rangle/\sqrt{1 - \varepsilon^2} = \varepsilon g_1/\sqrt{1 - \varepsilon^2}$$
$$= \varepsilon\sqrt{(1 - \|\tilde{\mathbf{g}}\|^2)/(1 - \varepsilon^2)} \geq \|\tilde{\mathbf{g}}\|. \quad (3)$$

If $\|\mathbf{g} - \mathbf{e}_1\| \geq 2\varepsilon$, we can get $g_1 \leq 1 - 2\varepsilon^2$, which means

$$\|\tilde{\mathbf{g}}\|^2 = g_1^2 \sum_{i=2}^n \alpha_i^2 = 1 - g_1^2$$
$$\geq 1 - (1 - 2\varepsilon^2)^2 = 4\varepsilon^2 - 4\varepsilon^4 \geq 2\epsilon^2, \quad (4)$$

and

$$\varepsilon\langle \mathbf{v}, \mathbf{g} \rangle/\sqrt{1 - \varepsilon^2} = \varepsilon\sqrt{(1 - \|\tilde{\mathbf{g}}\|^2)/(1 - \varepsilon^2)} \leq \frac{\|\tilde{\mathbf{g}}\|}{\sqrt{2}} \quad (5)$$

since $\varepsilon \leq 1/\sqrt{2}$.

Consider our task to determine whether $\|\mathbf{u}\| \leq \varepsilon$ or $\|\mathbf{u}\| \geq 2\varepsilon$. By Eq. (2) and Eq. (4), it suffices to determine whether $\|\tilde{\mathbf{g}}\| \leq \varepsilon$ or $\|\tilde{\mathbf{g}}\| \geq \sqrt{2}\varepsilon$. For any $\mathbf{y} \in S_{n-1}$, if we call Algorithm 1 with input $(\mathbf{x}, \alpha_{\mathbf{y}}/\|\alpha_{\mathbf{y}}\|, \Delta)$ and get $\langle \nabla f(\mathbf{x}), \alpha_{\mathbf{y}}/\|\alpha_{\mathbf{y}}\| \rangle \leq \Delta$, we have

$$\langle \mathbf{y}, \tilde{\mathbf{g}} \rangle = \langle (0, \mathbf{y}), \mathbf{g} \rangle \leq \frac{\varepsilon}{\sqrt{(n-1)(1-\varepsilon^2)}}\langle \mathbf{v}, \mathbf{g} \rangle + \Delta\|\alpha_{\mathbf{y}}\|$$
$$\leq \frac{\varepsilon}{\sqrt{(n-1)(1-\varepsilon^2)}}\langle \mathbf{v}, \mathbf{g} \rangle + \frac{\varepsilon}{25n}.$$

Otherwise, we have

$$\langle \mathbf{y}, \tilde{\mathbf{g}} \rangle \geq \frac{\varepsilon}{\sqrt{(n-1)(1-\varepsilon^2)}}\langle \mathbf{v}, \mathbf{g} \rangle - \frac{\varepsilon}{25n}.$$

Hence we can obtain

$$\Pr_{\mathbf{y}\sim S_{n-1}}\left[\langle\mathbf{y},\tilde{\mathbf{g}}\rangle\leq\frac{1}{\sqrt{n-1}}\cdot\frac{\varepsilon\langle\mathbf{v},\mathbf{g}\rangle}{\sqrt{1-\varepsilon^2}}-\frac{\varepsilon}{25n}\right]$$

$$\leq\Pr_{\mathbf{y}\sim S_{n-1}}\left[\text{Output ``}\langle\nabla f,\alpha_{\mathbf{y}}/\|\alpha_{\mathbf{y}}\|\rangle\leq\Delta\text{'' in Line 5}\right]$$

$$\leq\Pr_{\mathbf{y}\sim S_{n-1}}\left[\langle\mathbf{y},\tilde{\mathbf{g}}\rangle\leq\frac{1}{\sqrt{n-1}}\cdot\frac{\varepsilon\langle\mathbf{v},\mathbf{g}\rangle}{\sqrt{1-\varepsilon^2}}+\frac{\varepsilon}{25n}\right]$$

By Lemma 2.1, if $\|\mathbf{u}\|\leq\varepsilon$, which implies $\|\tilde{\mathbf{g}}\|\leq\varepsilon$ and Eq. (3), we can get

$$\mathbb{E}[N/T]\geq\Pr_{\mathbf{y}\sim S_{n-1}}\left[\langle\mathbf{y},\tilde{\mathbf{g}}\rangle\leq\frac{1}{\sqrt{n-1}}\cdot\frac{\varepsilon\langle\mathbf{v},\mathbf{g}\rangle}{\sqrt{1-\varepsilon^2}}-\frac{\varepsilon}{25n}\right]$$

$$\geq\Pr_{\mathbf{y}\sim S_{n-1}}\left[\langle\mathbf{y},\tilde{\mathbf{g}}\rangle\leq\frac{\|\tilde{\mathbf{g}}\|}{\sqrt{n-1}}-\frac{\|\tilde{\mathbf{g}}\|}{25(n-1)}\right]$$

$$\geq\frac{1}{2}+\frac{1}{2}\cdot\frac{3}{5}=\frac{4}{5}.$$

By Hoeffding inequality, we have

$$\Pr\left[\frac{4}{5}-\frac{N}{T}\geq\frac{1}{80}\right]\leq e^{-2T/80^2}\leq\delta=1/3,$$

which means, from Line 8, we at least have the probability of $2/3$ to get the right answer Yes. Otherwise, if $\|\mathbf{u}\|\geq2\varepsilon$, which implies $\|\tilde{\mathbf{g}}\|\geq\sqrt{2}\varepsilon$ and Eq. (5), we have

$$\mathbb{E}[N/T]\leq\Pr_{\mathbf{y}\sim S_{n-1}}\left[\langle\mathbf{y},\tilde{\mathbf{g}}\rangle\leq\frac{1}{\sqrt{n-1}}\cdot\frac{\varepsilon\langle\mathbf{v},\mathbf{g}\rangle}{\sqrt{1-\varepsilon^2}}+\frac{\varepsilon}{25n}\right]$$

$$\leq\Pr_{\mathbf{y}\sim S_{n-1}}\left[\langle\mathbf{y},\tilde{\mathbf{g}}\rangle\leq\frac{\|\tilde{\mathbf{g}}\|}{\sqrt{2(n-1)}}+\frac{\|\tilde{\mathbf{g}}\|}{25(n-1)}\right]$$

$$\leq\frac{1}{2}+\frac{1}{2}\cdot\frac{11}{20}=\frac{31}{40}.$$

Similarly by Hoeffding inequality, we have

$$\Pr\left[\frac{N}{T}-\frac{31}{40}\geq\frac{1}{80}\right]\leq e^{-2T/80^2}\leq\delta=1/3,$$

which means, from Line 10, we at least have the probability of $2/3$ to get the right answer No. $\quad\square$

Finally, we note that if we only allow classical deterministic algorithms, then gradient testing actually has a tight query complexity of $\Theta(n)$:

**Theorem 3.3.** *The classical deterministic query complexity for solving Problem 1.1 with queries to a comparison oracle* $O_f^{\mathrm{Comp}}$ *(1) is* $\Theta(n)$.

Specifically, given the gradient $\mathbf{g}$ and target direction $\mathbf{v}=\mathbf{e}_1$, our algorithm (details given in Appendix B.1) estimates the ratio $g_i/g_1=\langle\mathbf{g},\mathbf{e}_i\rangle/\langle\mathbf{g},\mathbf{v}\rangle$ for the remaining $n-1$ orthogonal directions $\mathbf{e}_i$ within a constant multiplicative factor. This is achieved via multiplicative search for a parameter $\beta$

such that $\mathbf{g}$ is nearly orthogonal to $\beta\mathbf{e}_1-\mathbf{e}_i$, ensuring that the total query complexity across all $n-1$ directions remains $O(n)$. Then we distinguish the two instances by verifying whether $\sum_{i=2}^n|g_i/g_1|^2$ satisfies the geometric constraint of the YES case. The $\Omega(n)$ lower bound proof (details given in Appendix B.2) builds a hard distribution where the unknown gradient is either exactly along a target direction or is slightly "tilted" toward one randomly chosen orthogonal direction. Any single query almost never detects this tilt, and gradient testing has low success probability unless $\Omega(n)$ queries are made.

### 3.3. Gradient Estimation

For gradient estimation, we achieve the following result:

**Theorem 3.4.** *Algorithm 3 solves Problem 1.2 with success probability at least* $\frac{2}{3}$ *using* $O(n\log\frac{1}{\varepsilon})$ *queries. Moreover, we can increase success probability to* $1-\eta$ *using* $O(n\log\frac{1}{\varepsilon}\log\frac{1}{\eta})$ *queries for any* $\eta\in(0,1)$.

The intuition of the algorithm can be summarized as follows. Let $\mathbf{g}=\nabla f/\|\nabla f\|$. To estimate the gradient direction $\mathbf{g}$ within error $\varepsilon$, a natural approach is to estimate each coordinate to relative accuracy $\varepsilon/\sqrt{n}$. In particular, we call Algorithm 1 with input direction $\alpha_j\mathbf{e}_i-\alpha_i\mathbf{e}_j$ to roughly compare between $\alpha_i/\alpha_j$ and $g_i/g_j$. Assuming without loss of generality that $g_1$ has the largest magnitude among all coordinates, we can then perform a binary search over $[-1,1]$ to estimate each ratio $g_i/g_1$ to accuracy $\varepsilon/\sqrt{n}$.

However, the approach above requires $\Theta(n\log(n/\varepsilon))$ queries. To improve upon this, we adopt an idea similar to that used in Algorithm 5, which fully exploits the upper bound on $\sum_{i=1}^n|g_i/g_1|^2$: the quantity $|g_i/g_1|$ cannot be large for all coordinates simultaneously. Accordingly, the algorithm first computes an upper bound on each quantity $\sqrt{n}\cdot|g_i/g_1|$, denoted by $\ell_i$, which is obtained in Lines 10–14. Then, for each coordinate $i$, we perform a binary search over the interval $[-\ell_i/\sqrt{n},\ell_i/\sqrt{n}]$ to estimate $g_i/g_1$. Due to $\sum_i\ell_i^2/n=O(1)$, our binary search cost $\sum_i\log\frac{\ell_i}{\varepsilon}$ can be reduced to $O(n\log\frac{1}{\varepsilon})$.

We note that the random frame does not need to be fully Haar distributed. The proof only uses the constant moment guarantee summarized in Theorem A.2; hence an orthogonal 2-design, or any sufficiently accurate approximation satisfying the same guarantee, can in principle replace the Haar-random frame.

*Proof.* In Line 3, if $\langle\mathbf{v}_i,\mathbf{g}\rangle<\Delta_1/\gamma$, we flip $\mathbf{v}_i$. Then

$$\langle\mathbf{v}_i,\mathbf{g}\rangle\geq-\Delta_1/\gamma\geq-1/n. \tag{6}$$

Under Eq. (6), we can promise that the vector $\mathbf{u}_1$ in Line 4 satisfies $\mathbb{E}(\langle\mathbf{u}_1,\mathbf{g}\rangle)\geq0.7$ (this is formally stated as Theorem A.2 with full proofs deferred to Appendix A.2). Define

**Algorithm 3** Classical Gradient Estimation

1: **Input:** accuracy $\varepsilon$, lower bound $\gamma$ on $\|\nabla f\|$
2: Set $\Delta_1 = \gamma/n$, $\Delta_2 = \varepsilon\gamma/400\sqrt{n}$ and define $w_i(\beta) = \beta\mathbf{e}_1 - \mathbf{e}_i$
3: Sample a Haar-random orthonormal frame $\{\mathbf{v}_1, \mathbf{v}_2, \ldots, \mathbf{v}_n\}$ in $\mathbb{R}^n$. For each $i = 1, 2, \ldots, n$, call Algorithm 1 with input $(\mathbf{y}, \mathbf{v}_i, \Delta_1)$. If $\langle \mathbf{v}_i, \nabla f \rangle < \Delta_1$, replace $\mathbf{v}_i$ by $-\mathbf{v}_i$
4: Compute $\mathbf{u}_1 = \frac{1}{\sqrt{n}}\sum_{i=1}^n \mathbf{v}_i$. Rotate the frame so that $\mathbf{u}_1$ becomes the first basis vector $\mathbf{e}_1$
5: For any $i = 2, 3, \ldots, n$, initialize all $\ell_i = 1$.
6: **for** $i = 2$ **to** $n$ **do**
7: $\quad$ Call Algorithm 1 with input $(\mathbf{x}, \mathbf{e}_i, \Delta_1)$
8: $\quad$ **if** $\langle \nabla f, \mathbf{e}_i \rangle < \Delta_1$ **then**
9: $\quad\quad$ $\mathbf{e}_i \leftarrow -\mathbf{e}_i$
10: $\quad$ **repeat**
11: $\quad\quad$ Set $\beta = \ell_i/\sqrt{n}$ and call Algorithm 1 with input $(\mathbf{x}, w_i(\beta)/\|w_i(\beta)\|, \Delta_2)$
12: $\quad\quad$ **if** $\langle \nabla f, w_i(\beta)/\|w_i(\beta)\| \rangle \leq \Delta_2$ **then**
13: $\quad\quad\quad$ $\ell_i \leftarrow 2\ell_i$
14: $\quad$ **until** $\ell_i$ no change
15: $\quad$ Set $\alpha_{i,1} = -\ell_i/\sqrt{n}$ and $\alpha_{i,2} = \ell_i/\sqrt{n}$
16: $\quad$ **repeat**
17: $\quad\quad$ Set $\alpha_i = (\alpha_{i,1} + \alpha_{i,1})/2$
18: $\quad\quad$ Call Algorithm 1, input $(\mathbf{x}, w_i(\alpha_i)/\|w_i(\alpha_i)\|, \Delta_2)$
19: $\quad\quad$ **if** $\langle \nabla f, w_i(\alpha_i)/\|w_i(\alpha_i)\| \rangle \leq \Delta_2$ **then**
20: $\quad\quad\quad$ Set $\alpha_{i,1} = \alpha_i$
21: $\quad\quad$ **else**
22: $\quad\quad\quad$ Set $\alpha_{i,2} = \alpha_i$
23: $\quad$ **until** $\alpha_{i,2} - \alpha_{i,1} < \varepsilon/4\sqrt{n}$
24: **Return:** $\mathbf{h} = (\sum_i \alpha_i \mathbf{u}_i)/\sqrt{\sum_i \alpha_i^2}$ where $\alpha_1 = 1$

---

$p = \Pr[\langle \mathbf{u}_1, \mathbf{g} \rangle \geq 1/10]$. Because of $\langle \mathbf{u}_1, \mathbf{g} \rangle \leq 1$, we can get $p \geq 2/3$ by Markov's inequality. In the rest of the proof, we consider the case that the event

$$g_1 = \langle \mathbf{u}_1, \mathbf{g} \rangle \geq 1/10 \tag{7}$$

holds. In Lines 10–14, we call Algorithm 1 with input direction $w_i(\ell_i/\sqrt{n})$, which can roughly judge the sign of $\ell_i/\sqrt{n} - g_i/g_1$ (similar to Eq. (8)). If the sign is negative, we multiply $\ell_i$ by 2. As a result, we obtain

$$\langle \nabla f, w_i(\ell_i/\sqrt{n}) \rangle \geq -\Delta_2,$$
$$\frac{\ell_i}{\sqrt{n}}g_1 - g_i \geq -\Delta_2 \cdot \sqrt{1 + \ell_i^2/n}/\gamma \geq -10\Delta_2/\gamma,$$

and similarly

$$\langle \nabla f, w_i(\ell_i/2\sqrt{n}) \rangle \leq \Delta_2$$
$$\frac{\ell_i}{2\sqrt{n}}g_1 - g_i \leq \Delta_2 \cdot \sqrt{1 + \ell_i^2/4n}/\gamma \leq 10\Delta_2/\gamma.$$

Using the value we set for $\Delta_2$, the error from Algorithm 1 satisfies

$$\frac{10\Delta_2}{\gamma \cdot g_1} \leq \frac{100\Delta_2}{\gamma} \leq \frac{\varepsilon}{4\sqrt{n}},$$

hence we have

$$|g_i/g_1| - \frac{\varepsilon}{4\sqrt{n}} \leq \ell_i/\sqrt{n} \leq 2|g_i/g_1| + \frac{\varepsilon}{2\sqrt{n}}.$$

As a result, in Lines 10–14, we promise to get the smallest $\ell_i$ which is a power of 2 and satisfies $\ell_i/\sqrt{n} > g_i/g_1$ up to a small error above. With Eq. (7) holding, for $\|g\| = 1$, we have $\sum_{i=2}^n g_i^2/g_1^2 \leq 99 = O(1)$. Consequently, to find all $\ell_i$, the number of queries we need is

$$\sum_{i=2}^n 2(1 + \log \ell_i) \leq \sum_{i=2}^n 2\ell_i \leq \sum_{i=2}^n 2\ell_i^2 = O(n).$$

In Lines 16–23, our binary search needs $O(\log \frac{\ell_i}{\varepsilon})$ iterations to get the accuracy $\varepsilon/4\sqrt{n}$. If we both have

$$\langle \nabla f, w_i(\alpha_{i,1}) \rangle \leq \Delta_2 \qquad \langle \nabla f, w_i(\alpha_{i,2}) \rangle \geq -\Delta_2,$$

we will then call Algorithm 1 with input $(y, w_i((\alpha_{i,1} + \alpha_{i,2})/2), \Delta_2)$, and finally we can get

$$|\alpha_i - |g_i/g_1|| \leq \varepsilon/(4\sqrt{n}) + \frac{\Delta_2\sqrt{1 + \ell_i^2/n}}{\gamma \cdot g_1} \leq \varepsilon/(2\sqrt{n}),$$

which means that

$$\|\mathbf{h} - \mathbf{g}\|^2 \leq 2|g_1|^2 \sum_{i=1}^n (\alpha_i - |g_i/g_1|)^2 \leq \varepsilon^2/2 < \varepsilon^2.$$

During binary search, we use

$$\sum_i \log\left(\frac{4\ell_i}{\varepsilon}\right) \leq n \log \frac{1}{\varepsilon} + \sum_i \log(4\ell_i) = O\left(n \log \frac{1}{\varepsilon}\right)$$

queries. Hence one base run succeeds with probability at least $2/3$ and uses $O(n \log \frac{1}{\varepsilon})$ queries.

Now consider the $m = \lceil 18\ln(1/\eta) \rceil$ independent base runs in the algorithm. Define $\mathbf{h}^{(r)}$ to be the output of the $r$-th iteration and set $\mathcal{N}_r = \{s \in \{1, \ldots, m\} : \|\mathbf{h}^{(r)} - \mathbf{h}^{(s)}\| \leq 2\varepsilon/3\}$. We finally return any $\mathbf{h}^{(r)}$ with $|\mathcal{N}_r| \geq m/2$ and prove that it satisfies our requirements with success probability at least $1 - \eta$.

Let $X_r$ be the indicator that the $r$-th candidate satisfies $\|\mathbf{h}^{(r)} - \mathbf{g}\| \leq \varepsilon_0$. Since $\Pr[X_r = 1] \geq 2/3$, Hoeffding's inequality gives

$$\Pr\left[\sum_{r=1}^m X_r \leq m/2\right] \leq \exp(-m/18) \leq \eta.$$

Conditioned on the complementary event, more than half of the candidates are good. Any two good candidates are

within $2\varepsilon_0 = 2\varepsilon/3$ of each other, so every good candidate satisfies the selection condition $|\mathcal{N}_r| \geq m/2$. Thus the algorithm can select at least one candidate. Moreover, for any selected candidate $\mathbf{h}^{(r)}$, the set $\mathcal{N}_r$ intersects the set of good candidates because both sets have size greater than $m/2$ or at least $m/2$ with a strict majority of good candidates. Therefore, for some good $\mathbf{h}^{(s)}$,

$$\|\mathbf{h}^{(r)}-\mathbf{g}\| \leq \|\mathbf{h}^{(r)}-\mathbf{h}^{(s)}\|+\|\mathbf{h}^{(s)}-\mathbf{g}\| \leq 2\varepsilon/3+\varepsilon/3 = \varepsilon.$$

The total query complexity is $m \cdot O(n \log \frac{1}{\varepsilon}) = O(n \log \frac{1}{\varepsilon} \log \frac{1}{\eta})$. $\qquad\square$

## 4. Quantum Algorithms

We further give a quantum algorithm for gradient estimation:

**Theorem 4.1.** *Algorithm 4 solves Problem 1.2 using* $O(\log \frac{n}{\varepsilon})$ *queries with probability at least* $\frac{8}{15} - 2\varepsilon$.

As introduced in Section 2.3, Jordan (2005) assumed quantum query access to $\langle \mathbf{y}, \nabla f \rangle$ for each $\mathbf{y}$ and then use QFT to estimate $\nabla f$. However, we cannot directly estimate $\langle \mathbf{y}, \mathbf{g} \rangle$ in our comparison setting. Instead, we choose a direction $\mathbf{e}_1$ and approximate $\langle \mathbf{y}, \mathbf{g} \rangle / \langle \mathbf{e}_1, \mathbf{g} \rangle$. To achieve this, we apply Algorithm 1 with input direction $\mathbf{v} = (k, \tilde{\mathbf{y}})$, where we assume $\mathbf{y} = (y_1, \tilde{\mathbf{y}})$ and apply binary search on $k$ such that $\mathbf{v}$ is roughly orthogonal to $\mathbf{g}$. This indicates that the overlap between $\tilde{\mathbf{y}}$ and $\mathbf{g}$ is about $-k\langle \mathbf{g}, \mathbf{e}_1 \rangle$. Then, we add $y_1$, obtaining $(y_1 - k)\langle \mathbf{e}_1, \mathbf{g} \rangle$ as $\langle \mathbf{y}, \mathbf{g} \rangle$, and kick this back to the phase. Finally, we apply QFT to get the estimation.

*Proof.* By Lemma 2.1, we have

$$\Pr_{\mathbf{v} \sim S_n} [|\langle \mathbf{v}, \mathbf{g} \rangle| \geq 1/(5\sqrt{n})] \geq 4/5.$$

Therefore, without loss of generality, we can assume $\langle \mathbf{v}, \mathbf{g} \rangle \geq 1/(5\sqrt{n})$, which means for any $\mathbf{y} \in M$,

$$|\langle \mathbf{g}, \mathbf{y} \rangle| \leq \sqrt{n} \leq 5n|\langle \mathbf{g}, \mathbf{v} \rangle|,$$

and we can know that there exists $k \in [-n, n]$ such that

$$\langle \mathbf{g}, (k, \tilde{\mathbf{y}}) \rangle = 0.$$

During our binary search in Lines 6–13, for each $\mathbf{y}$, define $\alpha_{\mathbf{y}}(k) = (k, \tilde{\mathbf{y}})$. If we already obtain

$$\langle \nabla f, \alpha_{\mathbf{y}}(k_1)/\|\alpha_{\mathbf{y}}(k_1)\| \rangle \leq \Delta$$
$$\langle \nabla f, \alpha_{\mathbf{y}}(k_2)/\|\alpha_{\mathbf{y}}(k_2)\| \rangle \geq -\Delta,$$

we will then call Algorithm 1 with input $(\mathbf{x}, \alpha_{\mathbf{y}}((k_1 + k_2)/2)/\|\alpha_{\mathbf{y}}((k_1 + k_2)/2)\|, \Delta)$.

Finally, after $O(\log \frac{n}{\varepsilon})$ iterations, we get

$$\langle \nabla f, (k, \tilde{\mathbf{y}}) \rangle \leq \|(k, \tilde{\mathbf{y}})\|\Delta \leq (5n + n)\Delta = 6n\Delta,$$
$$\langle \nabla f, (k + \varepsilon^2/(8\pi n^{1.5}), \tilde{\mathbf{y}}) \rangle \geq -6n\Delta,$$

**Algorithm 4** Quantum Gradient Estimation

1: **Input:** $\mathbf{x} \in \mathbb{R}^d$, accuracy $\varepsilon$, lower bound $\gamma$ on $\|\nabla f\|$
2: Set $\Delta = \gamma\varepsilon^2/(48\pi n^3)$
3: Randomly sample a unit vector $\mathbf{v}$, and rotate the frame so that $\mathbf{v}$ becomes the first basis vector $\mathbf{e}_1$
4: Initialize $|\psi\rangle = \frac{1}{\sqrt{N}} \sum_{\mathbf{y} \in M} |\mathbf{y}\rangle|\phi\rangle$, where $M = \{\mathbf{y}|y_i = 0, \frac{1}{t}, \frac{2}{t}, \ldots, 1\}$, $t = \lceil 10n^2/\varepsilon \rceil$, and $|\phi\rangle = \frac{1}{\sqrt{t^2+1}} \sum_{j=0}^{t^2} e^{-2\pi ij/t}|j\rangle$
5: Set $k_1 = -5n$, and $k_2 = 5n$
6: **repeat**
7:     Set $k = (k_1 + k_2)/2$
8:     For each $\mathbf{y}$, call Algorithm 1 with input $(\mathbf{x}, (k, \tilde{\mathbf{y}})/\|(k, \tilde{\mathbf{y}})\|, \Delta)$
9:     **if** $\langle \nabla f, (k, \tilde{\mathbf{y}})/\|(k, \tilde{\mathbf{y}})\| \rangle < \Delta$ **then**
10:       Set $k_1 = k$
11:     **else**
12:       Set $k_2 = k$
13: **until** $k_2 - k_1 < \varepsilon^2/(8\pi n^{1.5})$
14: Set $h(\mathbf{y}) = y_1 - k$, and output $\lfloor h(\mathbf{y}) \cdot t^2/\sqrt{n} \rfloor$ on the second register
15: Apply the quantum transformation $U$, which satisfies $U|\mathbf{y}\rangle = \frac{1}{\sqrt{N}} \sum_{\mathbf{z} \in M} e^{-2\pi i\langle \mathbf{z}, \mathbf{y} \rangle t}|\mathbf{z}\rangle$, on our first register.
16: **Return** the measurement of the first register.

which also means

$$\langle \mathbf{g}, \mathbf{y} \rangle \leq (y_1 - k)\langle \mathbf{g}, \mathbf{v} \rangle + 6n\Delta/\gamma,$$
$$\langle \mathbf{g}, \mathbf{y} \rangle \geq (y_1 - k - \varepsilon^2/(8\pi n^{1.5}))\langle \mathbf{g}, \mathbf{v} \rangle - 6n\Delta/\gamma.$$

And then we have

$$\left| h(\mathbf{y}) - \frac{\langle \mathbf{g}, \mathbf{y} \rangle}{\langle \mathbf{g}, \mathbf{v} \rangle} \right| \leq \varepsilon^2/(8\pi n^{1.5})+\frac{6n\Delta}{\gamma\langle \mathbf{g}, \mathbf{v} \rangle} \leq \varepsilon^2/(4\pi n^{1.5}),$$

which means

$$\left| e^{2\pi ih(\mathbf{y})\cdot \frac{t}{5\sqrt{n}}} - e^{2\pi i\cdot \langle \frac{\mathbf{g}}{\langle \mathbf{g}, \mathbf{v} \rangle}, \mathbf{y} \rangle \cdot \frac{t}{5\sqrt{n}}} \right| \leq |e^{i\varepsilon} - 1| \leq \varepsilon.$$

In Line 14, we can get the quantum state

$$|\psi\rangle = \frac{1}{\sqrt{N}} \sum_{\mathbf{y} \in M} e^{2\pi ih(\mathbf{y})\cdot \frac{t}{5\sqrt{n}}} |\mathbf{y}\rangle|\phi\rangle.$$

As a result, we get

$$\left\| |\psi\rangle - \frac{1}{\sqrt{N}} \sum_{\mathbf{y} \in M} e^{2\pi i\langle \frac{\mathbf{g}}{\langle \mathbf{g}, \mathbf{v} \rangle}, \mathbf{y} \rangle \cdot \frac{t}{5\sqrt{n}}} |\mathbf{y}\rangle \right\| \leq \varepsilon.$$

By Proposition C.1, we can output the result $\mathbf{z}$ after the transformation in Line 15 such that

$$\left\| \mathbf{z} - \frac{\mathbf{g}}{5\sqrt{n} \cdot \langle \mathbf{g}, \mathbf{v} \rangle} \right\| \leq 2n^{1.5}/t \leq \varepsilon/(10\sqrt{n})$$

with probability at least $2/3 - 2\varepsilon$. This further implies

$$\left\| \frac{\mathbf{z}}{\|\mathbf{z}\|} - \mathbf{g} \right\| \le 2\sin\theta \le 10\sqrt{n} \left\| \mathbf{z} - \frac{\mathbf{g}}{5\sqrt{n} \cdot \langle \mathbf{g}, \mathbf{v} \rangle} \right\| \le \varepsilon,$$

where $\theta = \arg(\frac{\mathbf{z}}{\|\mathbf{z}\|}, \mathbf{g})$. In all, we can succeed with probability at least $\frac{4}{5} \cdot (\frac{2}{3} - 2\varepsilon) \ge \frac{8}{15} - 2\varepsilon$. In addition, the success probability can also be boosted to $1 - \eta$ with any $\eta \in (0, 1)$ similar to the proof of Theorem 3.4. □

## 5. Numerical Experiments

We complement our theoretical results with numerical experiments that (i) validate the empirical query complexity of the proposed algorithms, and (ii) demonstrate that plugging our gradient direction estimator into a NGD optimizer yields convergence comparable to access to the exact normalized gradient, and better than existing comparison based baselines under matched per-iteration query budgets. All experiments are run on an Apple MacBook Air (M2, 2022) with 8-core Apple M2 chip and 16 GB memory.

**Setup.** We use three test functions: a general strongly-convex quadratic $f(\mathbf{x}) = \frac{1}{2}\mathbf{x}^\top U^\top \Lambda U \mathbf{x}$ with $U$ Haar-orthogonal and $\Lambda$ a diagonal matrix whose eigenvalues are evenly spread over $[1, 10]$; a sparse quadratic function in which the first $s = 10$ coordinates carry weights evenly spread over $[1, 10]$ and the remaining coordinates carry a vanishing weight; and the extended Rosenbrock function $f(\mathbf{x}) = \sum_{i=1}^{n-1} \left[ 100(x_{i+1} - x_i^2)^2 + (1 - x_i)^2 \right]$.

**Correctness of our algorithms.** We evaluate the three classical algorithms developed in this paper: the randomized $O(1)$ gradient testing algorithm (Algorithm 2), the deterministic $O(n)$ gradient testing algorithm (Algorithm 5), and the gradient estimation algorithm (Algorithm 3). For testing, we construct an extreme YES instance at distance $0.95\,\varepsilon$ from $\nabla f(\mathbf{x})/\|\nabla f(\mathbf{x})\|$ and an extreme NO instance at distance $2.05\,\varepsilon$, and run 100 independent trials for each $(n, \varepsilon)$ configuration with $n \in \{10, 50, 100\}$ and $\varepsilon = 0.2$. Both testing algorithms attain an empirical success probability of $100\%$ on every configuration, far exceeding the constant probability guarantee in our theoretical analysis. For estimation, Table 2 reports the dependence on dimension at $\varepsilon = 0.2$ across all three test functions, and Table 3 reports the dependence on the target precision $\varepsilon$ on the extended Rosenbrock function at $n = 100$. The estimator returns a unit vector whose error is below $\varepsilon$ in nearly $100\%$ of trials, with average error well below the target precision. This confirms that the worst-case query bound is not tight in typical instances.

**Empirical query complexity.** Theorem 3.4 predicts $Q = O(n \log(1/\varepsilon))$ comparison queries for our estimation algorithm. We verify this by sweeping a $10 \times 10$ grid of $(n, \varepsilon)$

*Table 2.* Estimation accuracy and average query count for Algorithm 3 across dimensions, $\varepsilon = 0.2$, 100 trials per cell. Success = fraction of trials with $\ell_2$ error below $\varepsilon$.

| Function | $n$ | Success | Avg. error | Avg. queries |
|---|---|---|---|---|
| Gen. Quadratic | 10 | 1.00 | 0.0068 | 85.3 |
| | 50 | 1.00 | 0.0071 | 461.0 |
| | 100 | 1.00 | 0.0071 | 929.6 |
| Sparse Quadratic | 10 | 1.00 | 0.0068 | 86.4 |
| | 50 | 1.00 | 0.0071 | 458.5 |
| | 100 | 1.00 | 0.0071 | 922.3 |
| Ext. Rosenbrock | 10 | 1.00 | 0.0068 | 85.4 |
| | 50 | 1.00 | 0.0072 | 459.8 |
| | 100 | 1.00 | 0.0072 | 925.7 |

*Table 3.* Estimation accuracy and average query count for Algorithm 3 on the extended Rosenbrock function at $n = 100$ as the target precision $\varepsilon$ varies, 100 trials per cell.

| $\varepsilon$ | Success | Avg. error | Max. error | Avg. queries |
|---|---|---|---|---|
| 0.20 | 1.00 | 0.0072 | 0.0081 | 925.7 |
| 0.10 | 1.00 | 0.0036 | 0.0041 | 1024.9 |
| 0.05 | 1.00 | 0.0018 | 0.0020 | 1124.5 |
| 0.01 | 1.00 | 0.00045 | 0.00052 | 1320.7 |

values on the extended Rosenbrock function with $\gamma = 0.05$ and 100 trials per cell. Figure 1a shows the joint dependence of the average query count on $n$ and $\ln(1/\varepsilon)$, following the bilinear model

$$Q \approx w_1\, n \ln(1/\varepsilon) + w_2\, n + w_3 \ln(1/\varepsilon) + b.$$

Fitting this model jointly across the grid gives $w_1 \approx 1.38$, $w_2 \approx 7.06$, $w_3 \approx -1.41$, $b \approx -5.28$ with $R^2 = 0.999$, in tight agreement with the theoretical scaling.

**Optimization with the estimated direction.** We feed the direction returned by Algorithm 3 into adaptive NGD (Levy, 2017) with step size $\eta_t = R/\sqrt{2t}$, $R = 5$, giving ADANGD+OURS. Baselines on the extended Rosenbrock function in $n = 100$ (averaged over 10 random initializations) are: IDEAL ADANGD, the noise-free reference using the true normalized gradient; ZO-SGD (Nesterov & Spokoiny, 2017), a two-point Gaussian smoothing estimator with $m = 100$ probes; ZO-RANKSGD (Tang et al., 2024), an $(m, k)$ ranking oracle with $m = k = 100$; SCOBO (Cai et al., 2022), $m = 100$ pairwise sign comparisons per iteration. We take $\varepsilon = 0.2$ for our estimator, where the cost is of the same order as the baselines. We also report a line-search variant ($\ell = 5$ shrinking trials, factor 0.1). The results are shown in Figure 1b and Figure 1c.

After 200 iterations ADANGD+OURS reaches objective 0.23 with line search and 2.6 without, essentially matching IDEAL ADANGD (0.23 and 3.1); while the other three baselines still remain over 10. At a comparable per-iteration

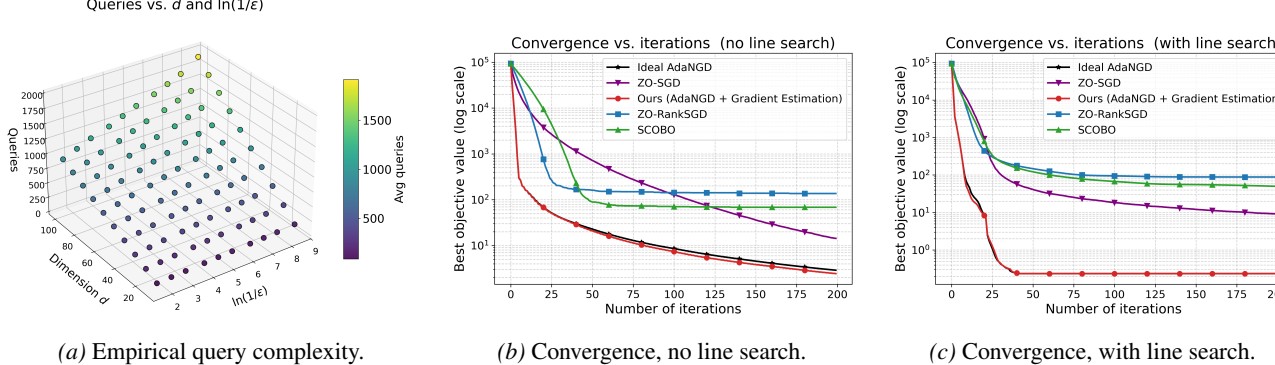

*(a)* Empirical query complexity.   *(b)* Convergence, no line search.   *(c)* Convergence, with line search.

*Figure 1.* (a) Average comparison queries used by Algorithm 3 on the extended Rosenbrock function over a $10 \times 10$ grid of $(n, \ln(1/\varepsilon))$ values; the fitted bilinear surface is $Q \approx 1.38\, n \ln(1/\varepsilon) + 7.06\, n - 1.41 \ln(1/\varepsilon) - 5.28$ with $R^2 = 0.999$. (b), (c) Convergence on the extended Rosenbrock function in $n = 100$, averaged over 10 random initializations, without and with line search respectively. Under per-iteration comparison budgets of the same order, ADANGD+OURS tracks the noise-free IDEAL ADANGD curve, while the baselines stall orders of magnitude above the optimum.

query budget, our estimator returns a direction whose angular deviation is small enough that adaptive NGD behaves as if it had exact gradient direction, whereas the other baselines spend the same budget on a much coarser direction.

## 6. Lower Bounds

We also establish lower bounds on gradient estimation, both in the classical and quantum settings.

**Theorem 6.1.** *Any classical algorithm that solves Problem 1.2 with success probability $2/3$ needs to take $\Omega(n \log \frac{1}{\varepsilon})$ comparison queries.*

The proof is built upon an $\varepsilon$-net argument on the sphere, and a gradient estimator must effectively identify which one is present. Each comparison query yields only a constant number of outcomes, conveys limited information, implying that the number of queries should scale linearly with dimension. The detailed proof can be found in Appendix D.1.

**Theorem 6.2.** *Any quantum algorithm that solves Problem 1.2 with success probability $2/3$ needs to take $\Omega(\log \frac{1}{\varepsilon})$ quantum comparison queries.*

This quantum lower bound argument applies the hybrid argument (Bennett et al., 1997), where we consider a hard instance with binary partial derivatives and prove that we need enough number of quantum queries to estimate the gradient. The detailed proof can be found in Appendix D.2.

## 7. Conclusions

In this paper, we study gradient testing and gradient estimation using comparison queries. For any smooth $f : \mathbb{R}^n \to \mathbb{R}$, $\mathbf{x} \in \mathbb{R}^n$, and $\varepsilon > 0$, our classical gradient testing algorithm determines whether $\nabla f(\mathbf{x})/\|\nabla f(\mathbf{x})\|$ is $\varepsilon$-close or $2\varepsilon$-far from a given unit vector $\mathbf{v}$ using $O(n)$ queries, as well as

a gradient estimation algorithm that outputs an $\varepsilon$-estimate of $\nabla f(\mathbf{x})/\|\nabla f(\mathbf{x})\|$ using $O(n \log(1/\varepsilon))$ queries. Both bounds are shown to be optimal. Furthermore, we study these problems in the quantum setting, and develop quantum algorithms for gradient testing and gradient estimation using $O(1)$ and $O(\log(n/\varepsilon))$ queries, respectively.

Technically, after an appropriate rotation, there exists a dominant coordinate whose inner product with the gradient captures a constant fraction of the gradient magnitude. We isolate this coordinate and express inner product as a scale of this constant fraction, and also optimized the application of binary search in other coordinates to achieve optimality. In terms of quantum algorithms, we leverage the quantum Fourier transform (QFT) to coherently encode directional derivatives across coordinates into relative phases of a single quantum state. This phase encoding enables the simultaneous extraction of relative inner products via inverse QFT.

Our work raises several open directions for future research. First, our quantum algorithm for gradient estimation is not yet optimal, and we conjecture that there exists an algorithm matching our lower bound. Second, it is also natural to study the gradient testing and estimation problems with a stochastic comparison oracle. Third, it would be valuable to investigate how the proposed gradient testing and estimation procedures can be incorporated into optimization solvers with improved performance.

## Acknowledgements

We thank the anonymous reviewers for their constructive feedback. XT, HW, YZ, and TL were supported by the National Natural Science Foundation of China (Grant Number 62372006).

## Impact Statement

This paper aims to advance machine learning research with a particular focus on optimization theory. The contributions are primarily theoretical and do not involve direct real-world applications. Therefore, we do not anticipate any immediate adverse societal consequences to be highlighted here.

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

# A. Auxiliary Lemmas for Classical Upper Bounds

## A.1. Inner Product Concentration for Random Vectors on Sphere

In this subsection, we give a formal proof of the inner product concentration for random vectors on sphere stated in Section 2.2, restated below:

**Lemma A.1.** *Let $n \geq 5$. For any $\mathbf{x} \in \mathbb{R}^n$, $\mathbf{x} \neq \mathbf{0}$, and any constant $c > 0$, there exists constant $p_1$ and $p_2$ which is independent of $n$, such that*

$$p_1 \leq \Pr_{\mathbf{y} \sim S_n} \left[ |\langle \mathbf{y}, \mathbf{x} \rangle| \leq \|\mathbf{x}\|/(c\sqrt{n}) \right] \leq p_2.$$

*where $\mathbf{y}$ is chosen from $S_n$ uniformly at random. In particular, we have the inequalities below:*

- $\Pr_{\mathbf{y} \sim S_n}[|\langle \mathbf{y}, \mathbf{x} \rangle| \leq \|\mathbf{x}\| \cdot \frac{24}{25\sqrt{n}}] \geq 3/5$

- $\Pr_{\mathbf{y} \sim S_n}[|\langle \mathbf{y}, \mathbf{x} \rangle| \leq \|\mathbf{x}\| \cdot \frac{18}{25\sqrt{n}}] \leq 11/20$

- $\Pr_{\mathbf{y} \sim S_n}[|\langle \mathbf{y}, \mathbf{x} \rangle| \geq \|\mathbf{x}\|/(5\sqrt{n})] \geq 4/5$

*Proof.* By rotational invariance of the uniform distribution on the sphere, there exists an orthogonal matrix $Q$ such that $Q\mathbf{x} = \|\mathbf{x}\|e_1$. Since $Q\mathbf{y} \sim \mathrm{Unif}(S_n)$ and $\langle \mathbf{y}, \mathbf{x} \rangle = \langle Q\mathbf{y}, Q\mathbf{x} \rangle$, we may assume without loss of generality that $\mathbf{x} = \|\mathbf{x}\|e_1$. Then

$$\frac{\langle \mathbf{y}, \mathbf{x} \rangle}{\|\mathbf{x}\|} = y_1.$$

Define the rescaled coordinate

$$T_n := \sqrt{n}\, y_1.$$

Thus,

$$\Pr\left( |\langle \mathbf{y}, \mathbf{x} \rangle| \leq \frac{\|\mathbf{x}\|}{c\sqrt{n}} \right) = \Pr\left( |T_n| \leq 1/c \right).$$

The marginal density of $y_1$ is

$$f_n(t) = c_n(1 - t^2)^{\frac{n-3}{2}}, \qquad t \in [-1, 1], \qquad c_n = \frac{\Gamma(n/2)}{\sqrt{\pi}\,\Gamma((n-1)/2)}.$$

By the change of variables $t = s/\sqrt{n}$, the random variable $T_n$ has density

$$h_n(s) = \frac{c_n}{\sqrt{n}} \left( 1 - \frac{s^2}{n} \right)^{\frac{n-3}{2}} \mathbf{1}_{\{|s| \leq \sqrt{n}\}}.$$

For $t \in [0, 1]$ define

$$F_n(t) := \Pr(|T_n| \leq t) = 2 \int_0^t h_n(s)\, \mathrm{d}s.$$

We claim that for every fixed $s \in [0, 1]$, the quantity $h_n(s)$ is nondecreasing in $n$ for all $n \geq 5$. Indeed, writing

$$h_n(s) = \frac{c_n}{\sqrt{n}} \left( 1 - \frac{s^2}{n} \right)^{\frac{n-3}{2}},$$

the factor $\frac{c_n}{\sqrt{n}}$ is nondecreasing in $n$ since, letting $z = (n-1)/2$,

$$\frac{c_n}{\sqrt{n}} = \frac{1}{\sqrt{\pi}} \frac{\Gamma(z + 1/2)}{\Gamma(z)} \frac{1}{\sqrt{2z + 1}},$$

and by Gautschi's inequality the ratio $\Gamma(z + 1/2)/\Gamma(z)$ grows at least on the order of $\sqrt{z}$. For the remaining factor, consider

$$\log\left( 1 - \frac{s^2}{n} \right)^{\frac{n-3}{2}} = \frac{n-3}{2} \log\left( 1 - \frac{s^2}{n} \right).$$

For $n \geq 5$ and $s \in [0, 1]$ we have $s^2/n \leq 1/5$. Differentiating with respect to $n$ and using the inequality $\log(1-u) \geq -\frac{u}{1-u}$ for $u \in (0, 1)$ shows that this expression is nondecreasing in $n$. Hence $h_n(s)$ is nondecreasing in $n$ for all $n \geq 5$. It follows that for each fixed $t \in [0, 1]$, the distribution function $F_n(t)$ is nondecreasing in $n$ for all $n \geq 5$.

Next, let $\mathbf{g} \sim \mathcal{N}(0, I_n)$ and define $\mathbf{y} = \mathbf{g}/\|\mathbf{g}\|$. Then $\mathbf{y} \sim \mathrm{Unif}(S_n)$ and

$$T_n = \frac{g_1}{\sqrt{\frac{1}{n} \sum_{i=1}^n g_i^2}}.$$

By the law of large numbers and Slutsky's theorem, $T_n$ converges in distribution to a standard normal random variable $Z \sim \mathcal{N}(0, 1)$. Consequently, for every fixed $t \geq 0$,

$$F_n(t) \longrightarrow 2\Phi(t) - 1,$$

where $\Phi$ denotes the standard normal cumulative distribution function.

We now verify the stated bounds. For $t = 0.96$, monotonicity yields $F_n(0.96) \geq F_5(0.96)$ for all $n \geq 5$. Since $f_5(u) = \frac{3}{4}(1 - u^2)$ on $[-1, 1]$, a direct computation gives

$$F_5(0.96) = 2 \int_0^{24/25} \frac{3}{4\sqrt{5}} \left(1 - \frac{s^2}{5}\right) \mathrm{d}s = \frac{105588}{78125\sqrt{5}} \geq \frac{3}{5},$$

which proves the first inequality.

For $t = 0.72$, monotonicity and the Gaussian limit imply

$$F_n(0.72) \leq \lim_{m \to \infty} F_m(0.72) = 2\Phi(0.72) - 1 < \frac{11}{20}, \qquad \forall n \geq 5,$$

establishing the second inequality.

Finally, for $t = 1/5$, the monotonicity of $F_n(1/5)$ implies that $\Pr(|T_n| \geq 1/5) = 1 - F_n(1/5)$ is non-increasing in $n$ for all $n \geq 5$. Using the Gaussian limit,

$$\Pr(|T_n| \geq 1/5) \geq \lim_{m \to \infty} \Pr(|T_m| \geq 1/5) = 2(1 - \Phi(1/5)) > \frac{4}{5},$$

which completes the proof. $\qquad\square$

## A.2. Constant Overlap via Basis Averaging

In this subsection, we prove that for a fixed unit vector in $\mathbb{R}^n$, assuming that its inner product with all basis vectors of a Haar-random orthonormal basis is at least $-1/n$, the given vector has a constant overlap with the uniform average of all basis vectors:

**Theorem A.2** (Constant Overlap via Basis Averaging)**.** *Fix $n \geq 3$ and a unit vector $\mathbf{v} \in \mathbb{R}^n$. Let $(\mathbf{u}_1, \ldots, \mathbf{u}_n)$ be a Haar-random orthonormal basis in $\mathbb{R}^n$, conditioned on the event*

$$E_n := \left\{ \langle \mathbf{u}_i, \mathbf{v} \rangle \geq -\frac{1}{n}, \ \forall i \in [n] \right\}.$$

*Define*

$$W_n := \left\langle \frac{1}{\sqrt{n}} \sum_{i=1}^n \mathbf{u}_i, \ \mathbf{v} \right\rangle.$$

*Then for all $n \geq 500$,*

$$\mathbb{E}[W_n \mid E_n] > 0.7.$$

*Proof.* Following the theorem statement, $(\mathbf{u}_1, \ldots, \mathbf{u}_n)$ is a Haar-random orthonormal basis. Set

$$X = (X_1, \ldots, X_n) := U^\top \mathbf{v} \sim \mathrm{Unif}(S_n), \qquad X_i = \langle \mathbf{u}_i, \mathbf{v} \rangle.$$

Then $W_n = \frac{1}{\sqrt{n}} \sum_{i=1}^{n} X_i$ and the conditioning event is $\mathsf{E}_n = \{X_i \geq -1/n \; \forall i\}$.

We use the Gaussian representation: let $G = (G_1, \ldots, G_n) \sim N(0, I_n)$ and $R := \|G\|/\sqrt{n}$. Then

$$X \stackrel{d}{=} \frac{G}{\|G\|} = \frac{1}{\sqrt{n}} \frac{G}{R}, \qquad \sqrt{n}\, X_i = \frac{G_i}{R}, \qquad W_n = \frac{\frac{1}{n} \sum_{i=1}^{n} G_i}{R}.$$

Moreover,

$$\mathsf{E}_n \text{ happens} \iff \frac{G_i}{R} \geq -\frac{1}{\sqrt{n}} \iff G_i \geq -\frac{R}{\sqrt{n}} \qquad (\forall i).$$

Since $R \to 1$ in probability and $R$ is sharply concentrated, this event is an $o(1)$ perturbation of $\{G_i \geq 0 \; \forall i\}$, and the conditional mean of $W_n$ converges to the $\alpha = 0$ limit $\sqrt{2/\pi}$.

We now make this quantitative with an explicit error term of order $1/\sqrt{n}$. Define $\alpha_n := 1/\sqrt{n}$. Let

$$m(\alpha) := \frac{\varphi(\alpha)}{\Phi(\alpha)}, \qquad \varphi(t) = \frac{1}{\sqrt{2\pi}} e^{-t^2/2}, \qquad \Phi(t) = \int_{-\infty}^{t} \varphi(s) \, ds.$$

A standard Taylor expansion at $\alpha = 0$ gives

$$m(\alpha) \geq \sqrt{\frac{2}{\pi}} - \frac{2}{\pi} \alpha \qquad \text{for all } \alpha \in [0, 1].$$

In particular, for all $n \geq 3$,

$$m(\alpha_n) \geq \sqrt{\frac{2}{\pi}} - \frac{2}{\pi} \frac{1}{\sqrt{n}}.$$

Next, to control the error introduced by replacing $R^{-1}$ with 1, recall that

$$R = \frac{\|G\|}{\sqrt{n}}, \qquad W_n = \frac{\frac{1}{n} \sum_{i=1}^{n} G_i}{R}.$$

We bound $\mathbb{E}\left[|R^{-1} - 1| \mid \mathsf{E}_n\right]$ explicitly.

Define the high-probability event

$$A := \{R \geq 3/4\}.$$

On the interval $[3/4, \infty)$, the function $x \mapsto x^{-1}$ has derivative $|x^{-2}| \leq (4/3)^2 = 16/9$, hence on $A$,

$$|R^{-1} - 1| \leq \frac{16}{9} |R - 1|.$$

Therefore,

$$\mathbb{E}\left[(|R^{-1} - 1|) \cdot \mathbf{1}_A \mid \mathsf{E}_n\right] \leq \frac{16}{9} \mathbb{E}\left[|R - 1| \mid \mathsf{E}_n\right].$$

Since $R = \sqrt{Y/n}$ with $Y = \|G\|^2 \sim \chi_n^2$, we have $\mathrm{Var}(R) \leq 1/n$ for all $n \geq 3$, and thus by the Cauchy-Schwarz inequality we have

$$\mathbb{E}[|R - 1|] \leq \sqrt{\mathrm{Var}(R)} \leq \frac{1}{\sqrt{2n}}.$$

Consequently,

$$\mathbb{E}\left[(|R^{-1} - 1|) \cdot \mathbf{1}_A \mid \mathsf{E}_n\right] \leq \frac{16}{9} \cdot \frac{1}{\sqrt{2n}}.$$

It remains to bound the contribution of the complement event $A^c = \{R < 3/4\}$. Since $R^2 = \|G\|^2/n$ and $\|G\|^2 \sim \chi_n^2$, a standard lower-tail bound gives

$$\Pr(A^c) = \Pr\left(\|G\|^2 \leq \frac{9}{16} n\right) \leq e^{-\frac{49}{1024} n}.$$

Moreover, for $n \geq 4$,

$$\mathbb{E}\left[\frac{1}{R^2}\right] = \frac{n}{n-2} \leq 2.$$

Hence, by the Cauchy-Schwarz inequality,

$$\mathbb{E}\left[(|R^{-1} - 1|) \cdot \mathbf{1}_{A^c}\right] \leq \mathbb{E}\left[\frac{1}{R}\mathbf{1}_{A^c}\right] + \Pr(A^c) \leq \sqrt{2\Pr(A^c)} + \Pr(A^c),$$

which is exponentially small in $n$ and in particular when $n \geq 500$, we can get

$$\mathbb{E}\left[(|R^{-1} - 1|) \cdot \mathbf{1}_{A^c}\right] \leq \frac{0.14}{\sqrt{n}}.$$

As a result, combining the two parts, we obtain

$$\mathbb{E}\left[|R^{-1} - 1| \mid \mathsf{E}_n\right] = \mathbb{E}\left[(|R^{-1} - 1|) \cdot \mathbf{1}_A \mid \mathsf{E}_n\right] + \mathbb{E}\left[(|R^{-1} - 1|) \cdot \mathbf{1}_{A^c}\right] \leq \frac{16}{9\sqrt{2n}} + \frac{0.14}{\sqrt{n}} < \frac{1.26}{\sqrt{n}} + \frac{0.14}{\sqrt{n}} = \frac{1.4}{\sqrt{n}}$$

which also means

$$\mathbb{E}[W_n \mid \mathsf{E}_n] \geq m(\alpha_n) - \frac{1.4}{\sqrt{n}} \qquad \text{for all } n \geq 3.$$

Combining the two inequalities,

$$\mathbb{E}[W_n \mid \mathsf{E}_n] \geq \sqrt{\frac{2}{\pi}} - \left(\frac{2}{\pi} + 1.4\right)\frac{1}{\sqrt{n}} \geq \sqrt{\frac{2}{\pi}} - \frac{2.04}{\sqrt{n}},$$

where we used $\frac{2}{\pi} + 1.4 < 2.04$.

Finally, if $n \geq 500$ then $2.04/\sqrt{n} \leq 0.0913$, hence

$$\mathbb{E}[W_n \mid \mathsf{E}_n] \geq 0.797884\ldots - 0.0913 > 0.7,$$

concluding the proof. $\qquad\square$

## B. Classical Deterministic Gradient Testing

In this section, prove the tight $\Theta(n)$ bound for classical deterministic gradient testing.

### B.1. Classical Deterministic Gradient Testing Algorithm

We first give a classical deterministic algorithm for gradient testing.

**Theorem B.1.** *Algorithm 5 solves Problem 1.1 using $O(n)$ queries to a comparison oracle $O_f^{\text{Comp}}$ in Eq. (1).*

Denote $\alpha_i = |g_i|/|g_1|$ for any $i = 2, 3, \ldots, n$, where $\mathbf{g}$ is the normalized gradient (see Section 2.1). Define $\delta = \sqrt{1/(1 - \varepsilon^2/2)^2 - 1}$. The distance $\|\mathbf{g} - \mathbf{v}\|$ captures angular deviation, while the ratios $\alpha_i$ capture relative coordinate imbalance. The following lemma establishes that, to solve Problem 1.1, it suffices for us to estimate each $\alpha_i$ within a constant multiplicative error.

**Lemma B.2.** *If $\|\mathbf{g} - \mathbf{v}\| \leq \varepsilon$, we have $\sum_{i=2}^n \alpha_i^2 \leq \delta^2$ and $g_1^2 \sum_{i=2}^n \alpha_i^2 \leq \varepsilon^2$. If $\|\mathbf{g} - \mathbf{v}\| \geq 2\varepsilon$, we have $\sum_{i=2}^n \alpha_i^2 \geq 4\delta^2$ and $g_1^2 \sum_{i=2}^n \alpha_i^2 \geq 2\varepsilon^2$.*

*Proof.* Due to $\mathbf{v} = \mathbf{e}_1 = (1, 0, \ldots, 0)$ and $\|\mathbf{g}\| = 1$, we have

$$\|\mathbf{g} - \mathbf{v}\|^2 = (1 - g_1)^2 + g_2^2 + \cdots + g_n^2 = (1 - g_1)^2 + 1 - g_1^2 = 2 - 2g_1.$$

If $\|\mathbf{g} - \mathbf{v}\| \leq \varepsilon$, we can get $g_1 \geq 1 - \varepsilon^2/2$, which means

$$\sum_{i=2}^n \alpha_i^2 = (1 - g_1^2)/g_1^2 \leq 1/(1 - \varepsilon^2/2)^2 - 1 \leq \delta^2.$$

---

**Algorithm 5** Classical Deterministic Gradient Testing

1: **Input:** A point $\mathbf{x} \in \mathbb{R}^n$, testing direction $\mathbf{v}$, accuracy $\varepsilon \in (0, 1/\sqrt{2})$, lower bound $\gamma$ on $\|\nabla f(\mathbf{x})\|$
2: **Output:** Test whether $\|\mathbf{g} - \mathbf{v}\| \leq \varepsilon$ (YES case) or $\|\mathbf{g} - \mathbf{v}\| \geq 2\varepsilon$ (NO case)
3: Set $\delta = \sqrt{1/(1 - \varepsilon^2/2)^2 - 1}$, $\Delta_1 = \gamma/(7n)$, $\Delta_2 = \frac{\gamma}{8n^2}$, $\Delta_3 = \frac{\gamma\delta}{30\sqrt{14}n^{1.5}}$ and define $w_i(\beta) := \beta\mathbf{e}_1 - \mathbf{e}_i$
4: For any $i = 2, 3, \ldots, n$ initialize all $\ell_i = 1$, $t_i = 1$
5: **for** $i = 2$ **to** $n$ **do**
6:    Call Algorithm 1 with input $(\mathbf{x}, \mathbf{e}_i, \Delta_1)$
7:    **if** $\langle \nabla f, \mathbf{e}_i \rangle < \Delta_1$ **then**
8:       $t_i \leftarrow -t_i$
9: Let $D = \mathrm{diag}(1, t_2, \ldots, t_n)$ and apply the orthogonal transformation $D$ to the coordinate frame.
10: Set $\mathbf{y} = (2n, -1, \ldots, -1)$ and call Algorithm 1 with input $(\mathbf{x}, \mathbf{y}/\|\mathbf{y}\|, \Delta_2)$
11: **if** $\langle \nabla f, \mathbf{y}/\|\mathbf{y}\| \rangle \leq \Delta_2$ **then**
12:    **Return** No
13: **for** $i = 2$ **to** $n$ **do**
14:    **loop**
15:       Set $\beta = \delta\ell_i/\sqrt{7n}$, and call Algorithm 1 with input $(\mathbf{x}, w_i(\beta)/\|w_i(\beta)\|, \Delta_3)$
16:       **if** $\langle \nabla f, w_i(\beta)/\|w_i(\beta)\| \rangle \leq \Delta_3$ **then**
17:          $\ell_i \leftarrow 3\ell_i/2$
18:       **else**
19:          **Break**
20:    **if** $\sum_{j=2}^{n} \ell_j^2 \geq 21n$ **then**
21:       **Break** and **Return** No
22: **Return** Yes

---

If $\|\mathbf{g} - \mathbf{v}\| \geq 2\varepsilon$, we can get $g_1 \leq 1 - 2\varepsilon^2$, which means

$$\sum_{i=2}^{n} \alpha_i^2 = (1 - g_1^2)/g_1^2 \geq 1/(1 - 2\varepsilon^2)^2 - 1 \geq 4\delta^2.$$

$\square$

Intuitively, Algorithm 5 proceeds as follows: We first choose a suitable $\mathbf{y}$ to exclude the case when $\langle \mathbf{v}, \mathbf{g} \rangle$ is too small, which implies $\mathbf{v}$ belongs to the NO case. Then we call Algorithm 1 and use $w_i(\beta) := \beta\mathbf{e}_1 - \mathbf{e}_i$ to compare $\beta$ and $|g_i/g_1|$ for each $i$. Note that if we call Algorithm 1 with input direction $w_i(\beta)$, we can roughly get the sign of

$$\langle \nabla f, w_i(\beta)/\|w_i(\beta)\| \rangle = \|\nabla f\|(\beta g_1 - g_i)/\|w_i(\beta)\|, \tag{8}$$

which is determined by $\beta - g_i/g_1$. Therefore we can roughly get the sign of $\beta - g_i/g_1$ by it.

If $\beta$ is roughly less than $|g_i/g_1|$, it is multiplied by $3/2$, and finally we find $\beta = \delta\ell_i/\sqrt{7n}$ which satisfies $2\beta/3 < |g_i/g_1| < \beta$ approximately. And then we can count $\sum_{i=2}^{n} \ell_i^2$ to get the range of $\|\mathbf{v} - \mathbf{g}\|$.

*Proof of Theorem B.1.* In Line 11, we exclude the extreme case where $g_1 = \langle \mathbf{g}, \mathbf{v} \rangle$ is very small. On the one hand, if

$$\langle \nabla f, \mathbf{y}/\|\mathbf{y}\| \rangle \leq \Delta,$$

we have

$$2ng_1 \leq \sum_{i=2}^{n} g_i + \frac{4n\Delta_2}{\gamma} \leq \sqrt{n\sum_{i=2}^{n} g_i^2 + \frac{1}{2n}} = \sqrt{n(1 - g_1^2)} + \frac{1}{2n} \leq \sqrt{n} + \frac{1}{2n},$$

which means

$$\langle \mathbf{v}, \mathbf{g} \rangle = g_1 \leq \frac{1}{\sqrt{n}}.$$

We can get that $\mathbf{v}$ belongs to the NO case. On the other hand, if

$$\langle \nabla f, \mathbf{y}/\|\mathbf{y}\| \rangle \geq -\Delta,$$

we have

$$2ng_1 \geq \sum_{i=2}^{n} g_i - \frac{4n\Delta_2}{\gamma} \geq \frac{1}{2} \sum_{i=2}^{n} g_i^2 - \frac{1}{2n} = \frac{1}{2}(1 - g_1^2) - \frac{1}{2n},$$

which means

$$g_1 \geq \frac{1}{10n}.$$

Therefore, in the rest of the proof, we assume

$$g_1 = \langle \mathbf{g}, \mathbf{v} \rangle \geq \frac{1}{10n}. \tag{9}$$

Combine Eq. (8), in our Lines 5–21, if we get $\delta\ell_i/\sqrt{7n} - |\alpha_i|$ is negative within error $\Delta_3$, we will multiply $\ell_i$ by $3/2$, and if $\ell_i$ remains unchanged, we promise that we find the smallest integer $\ell_i$ which is an integer power of $3/2$ and satisfies $\delta\ell_i/\sqrt{7n} \geq |\alpha_i|$ (up to $\Delta_3$, the precision of calling Algorithm 1 in Line 15). Consequently, for any $i = 2, 3, \ldots, n$, we can find such an $\ell_i$ using $\lceil 1 + \log \ell_i \rceil$ comparison queries.

On the one hand, we prove that if $\sum_{j=2}^{n} \ell_j^2 \geq 21n$, we will have $\|\mathbf{g} - \mathbf{v}\| \geq \varepsilon^2$. Denote $S = \{i \mid i \in \{2, 3, \ldots, n\}, \ell_i > 1\}$. Note that if $i \in S$, it means that we have started the recursive setting of powers of $3/2$ in Lines 5–21 above, and we can finally get $\ell_i$ which do not satisfy the judge in Line 16. It means that we have

$$\langle \nabla f, w_i(2\delta\ell_i/3\sqrt{7n})/\|w_i(2\delta\ell_i/3\sqrt{7n})\| \rangle \leq \Delta_3, \tag{10}$$

$$\langle \nabla f, w_i(\delta\ell_i/\sqrt{7n})/\|w_i(\delta\ell_i/\sqrt{7n})\| \rangle \geq -\Delta_3. \tag{11}$$

Combine Eq. (8) and Eq. (10), we can get

$$\frac{2\delta\ell_i}{3\sqrt{7n}}g_1 - g_i \leq \frac{\Delta_3}{\gamma}\sqrt{1 + \frac{4\delta^2\ell_i^2}{63n}} \leq \sqrt{2}\Delta_3/\gamma,$$

$$|\alpha_i| = |g_i/g_1| \geq \frac{2\delta\ell_i}{3\sqrt{7n}} - \sqrt{2}\Delta_3/(\gamma g_1).$$

By our choice of $\Delta_3$ in Line 3 and Eq. (9), we have

$$|\alpha_i| \geq \frac{2\delta\ell_i}{3\sqrt{7n}} - \frac{\delta}{3\sqrt{7n}}.$$

Applying the following Cauchy-Schwarz inequality,

$$\left(|\alpha_i|^2 + \frac{\delta^2}{9n}\right) \cdot \left(1 + \frac{1}{7}\right) \geq \left(|\alpha_i| + \frac{\delta}{3\sqrt{7n}}\right)^2,$$

we obtain

$$|\alpha_i|^2 \geq \frac{\delta^2\ell_i^2}{18n} - \frac{\delta^2}{9n}.$$

Therefore, if $\sum_{i=2}^{n} \ell_i^2 \geq 21n$, we have $\sum_{i \in S} \ell_i^2 \geq 20n$, and

$$\sum_{i \in S} |\alpha_i|^2 \geq \sum_{i \in S} \left(\frac{\delta^2\ell_i^2}{18n} - \frac{\delta^2}{9n}\right) \geq \delta^2 \left(\frac{20}{18} - \frac{1}{9}\right) = \delta^2.$$

Under the promise of Problem 1.1 and Lemma B.2, this implies that $\mathbf{x}$ belongs to the NO case.

On the other hand, consider the case where $\sum_{i=2}^{n} \ell_i^2 \leq 21n$. From Eq. (11), we have

$$|\alpha_i| \leq \frac{\ell_i}{\sqrt{7n}} + \sqrt{1 + \frac{\delta^2 \ell_i^2}{7n}} \Delta_3/(\gamma g_1).$$

Use the value of $\Delta_3$ we set in Line 3 and Eq. (9), we have

$$|\alpha_i| \leq \frac{\delta \ell_i}{\sqrt{7n}} + \frac{\delta}{3\sqrt{7n}}.$$

By the Cauchy-Schwarz inequality, we get

$$|\alpha_i|^2 \leq \left(1 + \frac{1}{6}\right) \left(\frac{\delta^2 \ell_i^2}{7n} + \frac{\delta^2}{63n}\right) \leq \frac{\delta^2 \ell_i^2}{6n} + \frac{\delta^2}{9n}.$$

If $\sum_{i=2}^{n} \ell_i^2 \leq 21n$, we have

$$\sum_{i=2}^{n} |\alpha_i|^2 \leq \sum_{i=2}^{n} \left(\frac{\delta^2 \ell_i^2}{6n} + \frac{\delta^2}{9n}\right) \leq \delta^2 \left(\frac{21}{6} + \frac{1}{9}\right) < 4\delta^2.$$

Under the promise of Problem 1.1 and Lemma B.2, this implies that $\mathbf{x}$ belongs to the YES case.

Next, we discuss the query complexity of Algorithm 5. Note that when Algorithm 5 terminates, all $\ell_i$ satisfy

$$\sum_{i=2}^{n} \ell_i \leq \sum_{i=2}^{n} \ell_i^2 \leq \frac{9}{4} \cdot \sum_{i=2}^{n} \max\{2\ell_i/3, 1\}^2 = O(n).$$

Based on our recursive setting in Lines 5–21, using

$$\sum_{i=1}^{n} (1 + \log \ell_i) \leq n + \sum_{i=1}^{n} \ell_i \tag{12}$$

queries, we can find all values of $\ell_i$ with $O(n)$ queries. $\qquad \square$

## B.2. Classical Deterministic Query Lower Bound on Gradient Testing

Here, we prove our classical deterministic lower bound for gradient testing. This implies that our algorithm above is optimal.

Our lower bound is obtained by considering the special case where the objective function $f$ is a hyperplane function, i.e., $f(\mathbf{x}) = \langle \mathbf{g}, \mathbf{x} \rangle + \mathbf{b}$. It satisfies that
$$f(\mathbf{x} + \mathbf{y}) = f(\mathbf{x}) + \langle \nabla f(\mathbf{x}), \mathbf{y} \rangle$$
and the gradient on any point is $\mathbf{g}$. Then our comparison query oracle becomes

$$O_f^{\text{comp}}(\mathbf{x}, \mathbf{x} + \mathbf{y}) = \text{sgn}(\langle \nabla f(\mathbf{x}), \mathbf{y} \rangle) = \text{sgn}(\mathbf{g}, \mathbf{y}). \tag{13}$$

**Theorem B.3** (Classical deterministic lower bound for gradient testing). *Fix $\varepsilon \in (0, 1/10)$ and let $\mathbf{v} \in \mathbb{R}^n$ be any unit vector. In the reduced oracle model where an unknown unit vector $\mathbf{g} \in \mathbb{R}^n$ is fixed and each query specifies $\mathbf{w} \in \mathbb{R}^n$ and returns*

$$O_{\mathbf{g}}(\mathbf{w}) = \text{sgn}(\langle \mathbf{g}, \mathbf{w} \rangle),$$

*any deterministic algorithm that always decides correctly between*

$$\text{(YES)} \quad \|\mathbf{g} - \mathbf{v}\|_2 \leq \varepsilon \qquad \text{and} \qquad \text{(NO)} \quad \|\mathbf{g} - \mathbf{v}\|_2 > 2\varepsilon$$

*must make at least $n - 1$ queries. In particular, the deterministic query complexity of Problem 1.1 is $\Omega(n)$.*

*Proof.* By applying an orthogonal transformation, we may assume $\mathbf{v} = \mathbf{e}_1$. Consider any deterministic algorithm $\mathcal{A}$ that makes $q$ adaptive queries $\mathbf{w}^{(1)}, \mathbf{w}^{(2)}, \ldots, \mathbf{w}^{(q)} \in \mathbb{R}^n$ to the oracle and then outputs YES or NO. Now we prove that if $q \leq n - 2$, then algorithm $\mathcal{A}$ must err in some cases.

We first fix the execution of $\mathcal{A}$ on the input vector

$$\mathbf{g}_Y := \mathbf{e}_1,$$

which clearly satisfies $\|\mathbf{g}_Y - \mathbf{e}_1\|_2 = 0 \leq \varepsilon$. Since $\mathcal{A}$ is deterministic, the entire interaction transcript on $g_Y$ uniquely determines the sequence of queries $\mathbf{w}^{(1)}, \ldots, \mathbf{w}^{(q)}$ that $\mathcal{A}$ issues along this execution path.

Let

$$W := \mathrm{span}\{\mathbf{w}^{(1)}, \mathbf{w}^{(2)}, \ldots, \mathbf{w}^{(q)}\}.$$

Then $\dim(W) \leq q$. Assume $q \leq n - 2$. We claim that there exists a unit vector $\mathbf{u}$ such that

$$\mathbf{u} \perp \mathbf{e}_1 \qquad \text{and} \qquad \mathbf{u} \perp W.$$

Indeed, $\mathbf{e}_1^\perp$ has dimension $n - 1$, and $W$ has dimension at most $q \leq n - 2$, so the subspace $\mathbf{e}_1^\perp \cap W^\perp$ has dimension at least $(n - 1) - q \geq 1$, hence we can find a unit vector $\mathbf{u} \in \mathbf{e}_1^\perp \cap W^\perp$.

Now choose a parameter $\delta := 3\varepsilon$ and define

$$\mathbf{g}_N := \frac{\mathbf{e}_1 + \delta\mathbf{u}}{\|\mathbf{e}_1 + \delta\mathbf{u}\|_2}.$$

Since $\mathbf{u} \perp \mathbf{e}_1$, we have $\|\mathbf{e}_1 + \delta\mathbf{u}\|_2 = \sqrt{1 + \delta^2}$ and

$$\langle \mathbf{g}_N, \mathbf{e}_1 \rangle = \frac{1}{\sqrt{1 + \delta^2}}.$$

A direct computation gives

$$\|\mathbf{g}_N - \mathbf{e}_1\|_2^2 = 2 - 2\langle \mathbf{g}_N, \mathbf{e}_1 \rangle = 2 - \frac{2}{\sqrt{1 + \delta^2}}.$$

For $\delta = 3\varepsilon < 1/3$, we can get

$$\|\mathbf{g}_N - \mathbf{e}_1\|_2^2 \geq \delta^2 \left(1 - \frac{3\delta^2}{4}\right) \geq \frac{3\delta^2}{4} > 4\varepsilon^2$$

so $\|\mathbf{g}_N - \mathbf{e}_1\|_2 > 2\varepsilon$. Therefore $\mathbf{g}_N$ satisfies the NO condition.

Next, it remains to show that the deterministic algorithm $\mathcal{A}$, which makes $q$ queries to the given oracle, behaves identically on $\mathbf{g}_Y$ and $\mathbf{g}_N$. We prove this by induction: for $t \leq q$, the $t$-th query of $\mathcal{A}$ on $\mathbf{g}_N$ is also $\mathbf{w}^{(t)}$, and the output obtained is the same as in the case of $\mathbf{g}_Y$. The base case $t = 1$ is obvious. For a general $t \leq q$, if the first $t - 1$ queries on $\mathbf{g}_N$ are exactly $\mathbf{w}^{(1)}, \ldots, \mathbf{w}^{(t-1)}$ and the outputs obtained are the same as in the case of $\mathbf{g}_Y$, then the next query of the deterministic algorithm $\mathcal{A}$ must also be consistent with that on $\mathbf{g}_Y$, namely $\mathbf{w}_t$. Using $\mathbf{u} \perp \mathbf{w}^{(t)}$, we have

$$\langle \mathbf{g}_N, \mathbf{w}^{(t)} \rangle = \left\langle \frac{\mathbf{e}_1 + \delta\mathbf{u}}{\sqrt{1 + \delta^2}}, \mathbf{w}^{(t)} \right\rangle = \frac{\langle \mathbf{e}_1, \mathbf{w}^{(t)} \rangle + \delta\langle \mathbf{u}, \mathbf{w}^{(t)} \rangle}{\sqrt{1 + \delta^2}} = \frac{\langle \mathbf{e}_1, \mathbf{w}^{(t)} \rangle}{\sqrt{1 + \delta^2}}.$$

Since $\sqrt{1 + \delta^2} > 0$, this shows

$$\mathrm{sgn}(\langle \mathbf{g}_N, \mathbf{w}^{(t)} \rangle) = \mathrm{sgn}(\langle \mathbf{e}_1, \mathbf{w}^{(t)} \rangle) = \mathrm{sgn}(\langle \mathbf{g}_Y, \mathbf{w}^{(t)} \rangle).$$

Thus for every $t \leq q$, the $t$-th query and the output are identical on $\mathbf{g}_Y$ and $\mathbf{g}_N$. Consequently, $\mathcal{A}$ follows the same execution path and produces the same final output on both inputs, but one input is a YES instance and the other is a NO instance. Therefore $\mathcal{A}$ must err on at least one of them.

We conclude that no deterministic algorithm can solve the problem with $q \leq n - 2$ queries. Equivalently, any deterministic correct algorithm needs at least $n - 1$ queries, i.e., $\Omega(n)$. $\qquad\square$

## C. Quantum Gradient Estimation

Here, we prove the following quantum gradient estimation algorithm adapted from (Jordan, 2005) estimates gradients to certain precision assuming an approximate quantum evaluation oracle:

**Proposition C.1** (Quantum gradient estimation). *Let $n \geq 2$ and $t \in \mathbb{Z}_+$. Define $T := t + 1$ and*

$$M := \left\{0, \frac{1}{t}, \frac{2}{t}, \ldots, 1\right\}^n, \qquad N := |M| = T^n.$$

*For any $\mathbf{x} \in [0,1]^n$, define the quantum state*

$$|\Phi_{\mathbf{x}}\rangle := \frac{1}{T^{n/2}} \sum_{\mathbf{y} \in \{0,1,\ldots,T-1\}^n} e^{2\pi i \langle \mathbf{y}, \mathbf{x}\rangle} |\mathbf{y}\rangle. \tag{14}$$

*Let $F_T$ denote the $T$-dimensional quantum Fourier transform over $\mathbb{Z}_T$, i.e., $F_T |j\rangle = \frac{1}{\sqrt{T}} \sum_{k=0}^{T-1} e^{-2\pi i j k} |k\rangle$. Let $F := F_T^{\otimes n}$.*

*Consider the following procedure: apply $F^{\dagger}$ to the input state, measure in the computational basis obtaining $K \in \{0, 1, \ldots, T-1\}^n$, and output $\hat{\mathbf{v}} := \frac{1}{t} K \in M$. Let*

$$m := \left\lceil 2 + \frac{3}{2} n \right\rceil. \tag{15}$$

*Then for every $\mathbf{x} \in [0,1]^n$,*

$$\Pr\left[\|\hat{\mathbf{v}} - \mathbf{x}\|_2 \leq \frac{\sqrt{n}\,(m+1)}{t}\right] \geq \frac{2}{3}. \tag{16}$$

*Moreover, if the actual input state $|\psi\rangle$ satisfies $\|\,|\psi\rangle - |\Phi_{\mathbf{x}}\rangle\,\|_2 \leq \varepsilon$, then the output satisfies*

$$\Pr\left[\|\hat{\mathbf{v}} - \mathbf{x}\|_2 \leq \frac{\sqrt{n}\,(m+1)}{t}\right] \geq \frac{2}{3} - 2\varepsilon. \tag{17}$$

*Proof.* We first analyze the ideal input $|\Phi_{\mathbf{x}}\rangle$. Since

$$e^{2\pi i \langle \mathbf{y}, \mathbf{x}\rangle} = \prod_{j=1}^{n} e^{2\pi i y_j x_j} \quad \text{and} \quad F = F_T^{\otimes n},$$

the state $|\Phi_{\mathbf{x}}\rangle$ decomposes as a tensor product across coordinates. Consequently, after applying $F^{\dagger}$, the measurement outcome $K = (K_1, \ldots, K_n)$ consists of independent coordinates, where each $K_j$ arises from measuring

$$|\phi_{\alpha}\rangle := \frac{1}{\sqrt{T}} \sum_{y=0}^{T-1} e^{2\pi i y \alpha} |y\rangle, \qquad \alpha = x_j.$$

Applying $F_T^{\dagger}$ to $|\phi_{\alpha}\rangle$, the amplitude of observing $k \in \{0, \ldots, T-1\}$ is

$$a_k = \langle k | F_T^{\dagger} | \phi_{\alpha}\rangle = \frac{1}{T} \sum_{y=0}^{T-1} e^{2\pi i y(\alpha - k/T)}.$$

Let $\delta_k := \alpha - k/T$. Summing the geometric series gives

$$a_k = \frac{1}{T} \cdot \frac{1 - e^{2\pi i T \delta_k}}{1 - e^{2\pi i \delta_k}} = \frac{1}{T} e^{\pi i (T-1)\delta_k} \frac{\sin(\pi T \delta_k)}{\sin(\pi \delta_k)}.$$

Hence

$$\Pr[K_j = k] = \frac{1}{T^2} \left(\frac{\sin(\pi T \delta_k)}{\sin(\pi \delta_k)}\right)^2.$$

Let $\theta := T\alpha$. Then $\delta_k = (\theta - k)/T$ and

$$\Pr[K_j = k] = \frac{1}{T^2}\left(\frac{\sin(\pi(\theta - k))}{\sin(\pi(\theta - k)/T)}\right)^2.$$

Using $|\sin(\pi z)| \leq 1$ and the inequality

$$|\sin(\pi u)| \geq 2|u| \quad \text{for } |u| \leq \tfrac{1}{2},$$

we obtain, for all $k$,

$$\Pr[K_j = k] \leq \frac{1}{4(\theta - k)^2}.$$

Therefore, for any integer $m \geq 2$,

$$\Pr\left[|K_j - \theta| \geq m\right] \leq \sum_{r=m}^{\infty} \frac{1}{2r^2} \leq \frac{1}{2(m-1)}. \tag{18}$$

Let $\theta_j := Tx_j$. By Eq. (18),

$$\Pr\left[|K_j - \theta_j| \geq m\right] \leq \frac{1}{2(m-1)}.$$

Applying the union bound,

$$\Pr\left[\exists j : |K_j - \theta_j| \geq m\right] \leq \frac{n}{2(m-1)}.$$

With the choice of Eq. (15), this probability is at most $1/3$, hence

$$\Pr\left[\forall j : |K_j - \theta_j| < m\right] \geq \frac{2}{3}.$$

On this event,

$$|\hat{v}_j - x_j| \leq \frac{m+1}{t},$$

and thus

$$\|\hat{\mathbf{v}} - \mathbf{x}\|_2 \leq \frac{\sqrt{n}\,(m+1)}{t}.$$

This proves Eq. (16).

Let $U := F^\dagger$. Since $U$ is unitary,

$$\left\|\, U|\psi\rangle - U|\Phi_{\mathbf{x}}\rangle \,\right\|_2 = \left\|\, |\psi\rangle - |\Phi_{\mathbf{x}}\rangle \,\right\|_2 \leq \varepsilon.$$

Let $P$ and $Q$ denote the measurement distributions obtained from $U|\psi\rangle$ and $U|\Phi_{\mathbf{x}}\rangle$, respectively. For any event $E$,

$$|P(E) - Q(E)| \leq \tfrac{1}{2}\left\|\, |\psi\rangle\langle\psi| - |\Phi_{\mathbf{x}}\rangle\langle\Phi_{\mathbf{x}}| \,\right\|_1 \leq \left\|\, |\psi\rangle - |\Phi_{\mathbf{x}}\rangle \,\right\|_2 \leq \varepsilon.$$

Let $E$ be the event

$$\|\hat{\mathbf{v}} - \mathbf{x}\|_2 \leq \frac{\sqrt{n}\,(m+1)}{t}.$$

Since $Q(E) \geq 2/3$, it follows that

$$P(E) \geq \frac{2}{3} - \varepsilon \geq \frac{2}{3} - 2\varepsilon,$$

which proves Eq. (17). $\qquad\square$

# D. Proof Details of Gradient Estimation Lower Bounds

## D.1. Classical Lower Bound

In this subsection, we prove the classical lower bound on gradient estimation (Theorem 6.1). Similarly, this is also obtained by considering to the special case in Appendix B.2 and our query oracle becomes Eq. (13).

Our lower bound relies on the following two lemmas:

**Lemma D.1** (Large $\varepsilon$-separated set on $S_n$). *Let $n \geq 1$ and $0 < \varepsilon \leq 1$. There exists a set $M \subset S_n$ such that*

$$\forall \mathbf{x} \neq \mathbf{y} \in M, \quad \|\mathbf{x} - \mathbf{y}\| > \varepsilon, \quad and \quad |M| \geq \frac{2}{\varepsilon^n}.$$

*In particular, $|M| = \Omega\left((1/\varepsilon)^n\right)$ with an explicit constant $2$.*

*Proof.* We construct $M$ greedily as a *maximal* $\varepsilon$-separated subset of $S_n$: start with $M = \emptyset$ and keep adding points $\mathbf{x} \in S_n$ with $\|\mathbf{x} - \mathbf{y}\| > \varepsilon$ for all $\mathbf{y} \in M$, until this is no longer possible.

By construction, $M$ is $\varepsilon$-separated. Maximality implies the covering property

$$S_n \subseteq \bigcup_{\mathbf{x} \in M} B(\mathbf{x}, \varepsilon), \qquad B(\mathbf{x}, \varepsilon) := \{\mathbf{z} \in S^n : \|\mathbf{z} - \mathbf{x}\| \leq \varepsilon\},$$

because otherwise a point outside the union could be added to $M$, contradicting maximality.

Hence, writing $\sigma_n(\cdot)$ for the surface area measure on $S_n$,

$$\sigma_n(S_n) \ \leq \ \sum_{\mathbf{x} \in M} \sigma_n(B(\mathbf{x}, \varepsilon)) \ \leq \ |M| \sup_{\mathbf{x} \in S_n} \sigma_n(B(\mathbf{x}, \varepsilon)).$$

So it suffices to show the following lemma: for every $\mathbf{x} \in S_n$ and $0 < \varepsilon \leq 1$,

$$\sigma_n(B(\mathbf{x}, \varepsilon)) \ \leq \ \frac{1}{2} \varepsilon^n \sigma_n(S_n). \tag{19}$$

Assuming Eq. (19) is true, we get

$$\sigma_n(S_n) \ \leq \ |M| \cdot \frac{1}{2} \varepsilon^n \sigma_n(S_n) \quad \Rightarrow \quad |M| \ \geq \ \frac{2}{\varepsilon^n},$$

as claimed.

In the rest of the proof, we prove Eq. (19). Fix $\mathbf{x} \in S_n$ and let $Z$ be uniform on $S_n$. Then

$$\frac{\sigma_n(B(\mathbf{x}, \varepsilon))}{\sigma_n(S^n)} \ = \ \Pr\left[\|Z - \mathbf{x}\| \leq \varepsilon\right].$$

Let $T := \langle Z, \mathbf{x} \rangle \in [-1, 1]$. Since $\|Z - \mathbf{x}\|^2 = 2 - 2\langle Z, \mathbf{x} \rangle$, the event $\|Z - \mathbf{x}\| \leq \varepsilon$ is equivalent to

$$T \ \geq \ 1 - \frac{\varepsilon^2}{2}.$$

The density of $T$ is

$$f_n(t) = c_n(1 - t^2)^{\frac{n-1}{2}}, \qquad c_n := \frac{\Gamma\left(\frac{n+1}{2}\right)}{\sqrt{\pi}\,\Gamma\left(\frac{n}{2}\right)}.$$

For $t \in [0, 1]$ we have $1 - t^2 = (1 - t)(1 + t) \leq 2(1 - t)$, so with $\delta := \varepsilon^2/2$,

$$
\begin{aligned}
\Pr\left[T \geq 1 - \delta\right] &= \int_{1-\delta}^{1} c_n (1 - t^2)^{\frac{n-1}{2}} \, dt \\
&\leq \int_{1-\delta}^{1} c_n \left(2(1 - t)\right)^{\frac{n-1}{2}} \, dt \\
&= c_n \, 2^{\frac{n-1}{2}} \int_{0}^{\delta} u^{\frac{n-1}{2}} \, du \qquad (u = 1 - t) \\
&= c_n \, 2^{\frac{n-1}{2}} \cdot \frac{\delta^{\frac{n+1}{2}}}{\frac{n+1}{2}} = \frac{c_n}{n+1} \varepsilon^{n+1}.
\end{aligned}
$$

Using the standard gamma-ratio bound

$$
c_n = \frac{\Gamma\left(\frac{n+1}{2}\right)}{\sqrt{\pi}\,\Gamma\left(\frac{n}{2}\right)} \leq \sqrt{\frac{n}{2}}
$$

$$
\frac{c_n}{n+1} \leq \frac{1}{2} \qquad (\forall n \geq 1),
$$

we obtain

$$
\Pr\left[\|Z - x\| \leq \varepsilon\right] = \Pr\left[T \geq 1 - \frac{\varepsilon^2}{2}\right] \leq \frac{1}{2}\varepsilon^{n+1} \leq \frac{1}{2}\varepsilon^{n} \qquad (\varepsilon \leq 1),
$$

which is exactly Eq. (19). This finishes the proof. $\qquad\qquad\square$

We need another lemma on adaptive query lower bounds. We prove that any query model with three possible outcomes, which means $<, >, =$ in our comparison query, need to use at least $\log n$ queries to distinguish $n$ different results, no matter it is deterministic algorithm or random algorithm.

**Lemma D.2.** *Let $I$ be an unknown index uniformly distributed over $[n] := \{1, 2, \ldots, n\}$. Consider a query model in which each query returns one of three possible outcomes, and each $I$ corresponds to a type of query oracle. For different $I$, their oracles are also different. Any (possibly randomized) algorithm that identifies $I$ with success probability at least $2/3$ requires $\Omega(\log n)$ queries in the worst case.*

*Proof.* Consider the uniform distribution over $I$. By Yao's minimax principle, there exists a fixing of the internal randomness of $\mathcal{A}$ such that the resulting deterministic algorithm $\mathcal{A}_r$ succeeds with probability at least $2/3$ when $I$ is drawn uniformly from $[n]$. Therefore, it suffices to prove that any deterministic algorithm that makes at most $T$ queries has success probability at most $O(3^T/n)$ under the uniform distribution.

Now consider an arbitrary deterministic algorithm making at most $T$ adaptive queries. For such a determined algorithm, it can be represented as a ternary decision tree of depth at most $T$. Each internal node corresponds to a query, and each node has exactly three outgoing edges, one for each possible query outcome. Each leaf of the tree outputs a value in $[n]$.

A decision tree of depth at most $T$ has at most $3^T$ leaves, and hence the algorithm can correctly identify at most $3^T$ values of $I$. Under the uniform distribution over $I$, the success probability of any such deterministic algorithm is therefore at most $3^T/n$. To ensure that the success probability is least $2/3$, we must have $T \geq \log_3 n - O(1)$. This proves that any randomized algorithm identifying $I$ with constant success probability requires $\Omega(\log n)$ queries. $\qquad\square$

*Proof of Theorem 6.1.* By Lemma D.1, we can get an $\varepsilon$-net $M$ on $S_n$, which satisfies that for each $\mathbf{x}, \mathbf{y} \in M$, we have $\|\mathbf{x} - \mathbf{y}\| > \varepsilon$ and $|M| = \Omega((1/\varepsilon)^n)$.

Assume our gradient direction belongs to $M$. Note that each comparison query only has three types of possible responses. As a result, by Lemma D.2, if we want to distinguish each gradient direction in $M$, the query complexity is at least

$$
\log_3 |M| = \Omega(n \log(1/\varepsilon)).
$$

$\qquad\qquad\square$

## D.2. Quantum Lower Bound

In this subsection, we prove the quantum lower bound on gradient estimation (Theorem 6.2). It is still obtained by considering to the special case in Appendix B.2 and our quantum oracle becomes

$$O_{f,Q}^{\text{comp}}|\mathbf{x}\rangle|\mathbf{x}+\mathbf{y}\rangle|z\rangle = |\mathbf{x}\rangle|\mathbf{y}\rangle|z \oplus \text{sgn}(\langle \nabla f(\mathbf{x}), \mathbf{y}\rangle)\rangle. \tag{20}$$

Our proof is decomposed into three parts: a bound on average sensitivity of halfspaces (Appendix D.2.1), a bound on oracle distinction (Appendix D.2.2), and finally the proof of Theorem 6.2 (Appendix D.2.3).

### D.2.1. AVERAGE SENSITIVITY OF HALFSPACES

**Lemma D.3** (Average $k$-flip sensitivity of halfspaces). *There exists a universal constant $C' > 0$ such that the following holds. Let*

$$h(\mathbf{z}) = \mathbf{1}\left[\sum_{i=1}^{n} w_i z_i \geq 0\right], \qquad \mathbf{z} \in \{-1, 1\}^n,$$

*be a halfspace. Let $Z$ be uniform on $\{-1, 1\}^n$, and let $S$ be uniform over all $k$-subsets of $[n]$. Then*

$$\mathbb{E}_{S:\,|S|=k} \Pr\left[h(Z) \neq h(Z^{\oplus S})\right] \leq C'\sqrt{\frac{k}{n}}.$$

*Moreover, for all $k \leq n/2$ one may take $C'$ universal (independent of $n, k$).*

*Proof.* Scaling $\mathbf{w}$ by any $\lambda > 0$ does not change $h$, hence we may assume

$$\|\mathbf{w}\|_2^2 = \sum_{i=1}^{n} w_i^2 = 1.$$

Also, flipping the sign of a coordinate $z_i$ and simultaneously flipping the sign of $w_i$ leaves the distribution of $Z$ invariant and does not change the probability of disagreement. Thus, without loss of generality we may assume $w_i \geq 0$ for all $i$.

Then we make the decomposition for a fixed set $S$. Fix $S \subseteq [n]$ with $|S| = k$. Write

$$X := \sum_{i=1}^{n} w_i Z_i, \qquad B_S := \sum_{i \in S} w_i Z_i, \qquad Y_S := \sum_{i \notin S} w_i Z_i.$$

Then $X = Y_S + B_S$. Flipping the coordinates in $S$ sends $Z_i \mapsto -Z_i$ for $i \in S$, hence the corresponding linear form becomes

$$X' := \sum_{i=1}^{n} w_i (Z^{\oplus S})_i = Y_S - B_S.$$

Therefore,
$$h(Z) \neq h(Z^{\oplus S}) \iff \mathbf{1}[X \geq 0] \neq \mathbf{1}[X' \geq 0] \iff (X)(X') < 0 \text{ or } XX' = 0.$$

Since $XX' = (Y_S + B_S)(Y_S - B_S) = Y_S^2 - B_S^2$, we obtain the inclusion

$$\{h(Z) \neq h(Z^{\oplus S})\} \subseteq \{|Y_S| \leq |B_S|\}. \tag{21}$$

Let

$$\sigma_S^2 := \text{Var}(Y_S) = \sum_{i \notin S} w_i^2 = 1 - \sum_{i \in S} w_i^2 = 1 - \|w_S\|_2^2.$$

Consider the normalized sum $\widetilde{Y}_S := Y_S/\sigma_S$, which has variance 1. A standard anti-concentration inequality for Rademacher sums (a consequence of a local limit theorem / Berry-Esseen) states that there exists a universal constant $C_0 > 0$ such that for every $t > 0$,

$$\Pr\left(|\widetilde{Y}_S| \leq t\right) \leq C_0 t. \tag{22}$$

Equivalently, for every $u > 0$,

$$\Pr\left(|Y_S| \leq u\right) \ \leq \ C_0 \, \frac{u}{\sigma_S}. \tag{23}$$

Since the random variables $Y_S$ and $B_S$ depend on disjoint sets of coordinates of $Z$, we can get that $Y_S$ and $B_S$ are independent. Using Eq. (21), independence, and conditioning on $B_S$, we have

$$
\begin{aligned}
\Pr\left(h(Z) \neq h(Z^{\oplus S})\right) &\leq \Pr\left(|Y_S| \leq |B_S|\right) \\
&= \mathbb{E}\Big[\Pr\left(|Y_S| \leq |B_S| \mid B_S\right)\Big] \\
&\leq \mathbb{E}\Big[C_0 \, \frac{|B_S|}{\sigma_S}\Big] \qquad \text{(by Eq. (23) with } u = |B_S|) \\
&= \frac{C_0}{\sigma_S} \, \mathbb{E}|B_S|.
\end{aligned}
$$

By the Cauchy-Schwarz inequality, there exists a universal constant $C_1 > 0$ such that

$$\mathbb{E}|B_S| = \mathbb{E}\left|\sum_{i \in S} w_i Z_i\right| \leq C_1 \left(\sum_{i \in S} w_i^2\right)^{1/2} = C_1 \, \|w_S\|_2.$$

Hence

$$\Pr\left(h(Z) \neq h(Z^{\oplus S})\right) \ \leq \ C_0 C_1 \, \frac{\|w_S\|_2}{\sqrt{1 - \|w_S\|_2^2}}. \tag{24}$$

Define $a := \|w_S\|_2 \in [0, 1]$. We claim there is a universal $C_2 > 0$ such that for all $a \in [0, 1]$,

$$\min\left\{1, \, \frac{a}{\sqrt{1 - a^2}}\right\} \ \leq \ C_2 \, a. \tag{25}$$

Indeed, if $a \leq 1/2$ then $\sqrt{1 - a^2} \geq \sqrt{3}/2$ and $\frac{a}{\sqrt{1-a^2}} \leq \frac{2}{\sqrt{3}} a$. If $a > 1/2$, then $\min\{1, \frac{a}{\sqrt{1-a^2}}\} \leq 1 \leq 2a$. Thus Eq. (25) holds with $C_2 := 2$.

Combining Eq. (24) with the trivial bound $\Pr(h(Z) \neq h(Z^{\oplus S})) \leq 1$ and using Eq. (25), we get a universal constant $C_3 > 0$ such that for every fixed $S$,

$$\Pr\left(h(Z) \neq h(Z^{\oplus S})\right) \ \leq \ C_3 \, \|w_S\|_2. \tag{26}$$

Take the expectation over uniform $S$ with $|S| = k$:

$$\mathbb{E}_S \Pr\left(h(Z) \neq h(Z^{\oplus S})\right) \ \leq \ C_3 \, \mathbb{E}_S \|w_S\|_2.$$

By Jensen's inequality,

$$\mathbb{E}_S \|w_S\|_2 \ \leq \ \sqrt{\mathbb{E}_S \|w_S\|_2^2}.$$

But

$$\mathbb{E}_S \|w_S\|_2^2 = \mathbb{E}_S \sum_{i \in S} w_i^2 = \sum_{i=1}^{n} w_i^2 \, \Pr(i \in S) = \sum_{i=1}^{n} w_i^2 \, \frac{k}{n} = \frac{k}{n} \|w\|_2^2 = \frac{k}{n}.$$

Therefore,

$$\mathbb{E}_{S: \, |S|=k} \Pr\left[h(Z) \neq h(Z^{\oplus S})\right] \ \leq \ C_3 \sqrt{\frac{k}{n}}.$$

Setting $C' := C_3$ completes the proof. $\qquad\square$

D.2.2. DISTINCTION BOUND

**Lemma D.4.** *Let* $\mathbf{z} \in \{-1,1\}^n$ *and consider the oracle*

$$O_{\mathbf{z}} \ : \ |\mathbf{x}, \mathbf{y}\rangle \mapsto |\mathbf{x}, \mathbf{y} \oplus f(\mathbf{x}, \mathbf{z})\rangle,$$

*where* $\mathbf{x} \in \mathbb{Z}^n$ *(coordinates not necessarily $\pm 1$) and*

$$f(\mathbf{x}, \mathbf{z}) = \begin{cases} 1 & \text{if } \langle \mathbf{x}, \mathbf{z} \rangle \geq 0, \\ 0 & \text{if } \langle \mathbf{x}, \mathbf{z} \rangle < 0. \end{cases}$$

*Let*

$$M := \{(\mathbf{z}, \tilde{\mathbf{z}}) \in \{-1,1\}^n \times \{-1,1\}^n : H(\mathbf{z}, \tilde{\mathbf{z}}) = k\},$$

*where $H$ denotes the Hamming distance. Then the value*

$$S_1 := \sum_{(\mathbf{z}, \tilde{\mathbf{z}}) \in M} \left\| O_{\mathbf{z}} |\psi\rangle - O_{\tilde{\mathbf{z}}} |\psi\rangle \right\|^2$$

*for a normalized state $|\psi\rangle$ satisfies*

$$S_1 = O\left( \sqrt{\frac{k}{n}} \binom{n}{k} 2^n \right).$$

*Proof.* Let the Hilbert space be spanned by $\{|\mathbf{x}, \mathbf{y}\rangle\}$, where $\mathbf{x}$ is the query register and $\mathbf{y}$ is the answer register. Any normalized state can be written as

$$|\psi\rangle \ = \ \sum_{\mathbf{x}} |\mathbf{x}\rangle |\psi_{\mathbf{x}}\rangle, \qquad \sum_{\mathbf{x}} \||\psi_{\mathbf{x}}\rangle\|^2 = 1,$$

where $|\psi_{\mathbf{x}}\rangle$ is the (unnormalized) state on the second register conditional on query $\mathbf{x}$.

For each fixed $\mathbf{x}$ and $\mathbf{z}$, define the unitary on the answer register

$$U_{\mathbf{z}}^{(\mathbf{x})} : |\mathbf{y}\rangle \mapsto |\mathbf{y} \oplus f(\mathbf{x}, \mathbf{z})\rangle.$$

Then

$$O_{\mathbf{z}} |\psi\rangle - O_{\tilde{\mathbf{z}}} |\psi\rangle \ = \ \sum_{\mathbf{x}} |\mathbf{x}\rangle \big( U_{\mathbf{z}}^{(\mathbf{x})} - U_{\tilde{\mathbf{z}}}^{(\mathbf{x})} \big) |\psi_{\mathbf{x}}\rangle.$$

Let

$$D(\mathbf{z}, \tilde{\mathbf{z}}) \ := \ \{\mathbf{x} : f(\mathbf{x}, \mathbf{z}) \neq f(\mathbf{x}, \tilde{\mathbf{z}})\}.$$

If $\mathbf{x} \notin D(\mathbf{z}, \tilde{\mathbf{z}})$ then $U_{\mathbf{z}}^{(\mathbf{x})} = U_{\tilde{\mathbf{z}}}^{(\mathbf{x})}$ and that term vanishes. Therefore

$$\left\| O_{\mathbf{z}} |\psi\rangle - O_{\tilde{\mathbf{z}}} |\psi\rangle \right\|^2 = \left\| \sum_{\mathbf{x} \in D(\mathbf{z}, \tilde{\mathbf{z}})} |\mathbf{x}\rangle \big( U_{\mathbf{z}}^{(\mathbf{x})} - U_{\tilde{\mathbf{z}}}^{(\mathbf{x})} \big) |\psi_{\mathbf{x}}\rangle \right\|^2 = \sum_{\mathbf{x} \in D(\mathbf{z}, \tilde{\mathbf{z}})} \left\| \big( U_{\mathbf{z}}^{(\mathbf{x})} - U_{\tilde{\mathbf{z}}}^{(\mathbf{x})} \big) |\psi_{\mathbf{x}}\rangle \right\|^2,$$

Now, for any two unitaries $U, V$ and any vector $|\mathbf{v}\rangle$,

$$\|(U - V)|\mathbf{v}\rangle\| \ \leq \ \|U|\mathbf{v}\rangle\| + \|V|\mathbf{v}\rangle\| \ = \ 2\|\mathbf{v}\|.$$

Hence,

$$\left\| \big( U_{\mathbf{z}}^{(\mathbf{x})} - U_{\tilde{\mathbf{z}}}^{(\mathbf{x})} \big) |\psi_{\mathbf{x}}\rangle \right\|^2 \ \leq \ 4 \||\psi_{\mathbf{x}}\rangle\|^2, \quad \text{whenever } f(\mathbf{x}, \mathbf{z}) \neq f(\mathbf{x}, \tilde{\mathbf{z}}).$$

We conclude that for any $\mathbf{z}, \tilde{\mathbf{z}}$,

$$\left\| O_{\mathbf{z}} |\psi\rangle - O_{\tilde{\mathbf{z}}} |\psi\rangle \right\|^2 \ \leq \ 4 \sum_{\mathbf{x} : f(\mathbf{x}, \mathbf{z}) \neq f(\mathbf{x}, \tilde{\mathbf{z}})} \||\psi_{\mathbf{x}}\rangle\|^2. \tag{27}$$

Using Eq. (27),

$$S_1 \leq 4 \sum_{(\mathbf{z},\tilde{\mathbf{z}}) \in M} \sum_{\mathbf{x}: f(\mathbf{x},\mathbf{z}) \neq f(\mathbf{x},\tilde{\mathbf{z}})} \||\psi_{\mathbf{x}}\rangle\|^2$$

$$= 4 \sum_{\mathbf{x}} \||\psi_{\mathbf{x}}\rangle\|^2 \underbrace{\left| \{(\mathbf{z}, \tilde{\mathbf{z}}) \in M : f(\mathbf{x}, \mathbf{z}) \neq f(\mathbf{x}, \tilde{\mathbf{z}})\} \right|}_{=:N_{\mathbf{x}}}.$$

Therefore,

$$S_1 \leq 4 \sum_x \||\psi_{\mathbf{x}}\rangle\|^2 N_{\mathbf{x}} \leq 4 \left( \max_{\mathbf{x}} N_{\mathbf{x}} \right) \sum_{\mathbf{x}} \||\psi_{\mathbf{x}}\rangle\|^2 = 4 \max_{\mathbf{x}} N_{\mathbf{x}},$$

using $\sum_{\mathbf{x}} \||\psi_{\mathbf{x}}\rangle\|^2 = 1$. Hence, the problem reduces to bounding $N_{\mathbf{x}}$, where for fixed $\mathbf{x} \in \mathbb{Z}^n$ we define

$$g_{\mathbf{x}}(\mathbf{z}) := f(\mathbf{x}, \mathbf{z}) = \mathbf{1}[\langle \mathbf{x}, \mathbf{z} \rangle \geq 0], \quad \mathbf{z} \in \{-1, 1\}^n,$$

and

$$N_{\mathbf{x}} = |\{(\mathbf{z}, \tilde{\mathbf{z}}) \in M : g_{\mathbf{x}}(\mathbf{z}) \neq g_{\mathbf{x}}(\tilde{\mathbf{z}})\}|.$$

Note that the function $g_{\mathbf{x}}$ is a *halfspace* (linear threshold function) on the hypercube:

$$g_{\mathbf{x}}(\mathbf{z}) = \mathbf{1}\left[ \sum_{i=1}^{n} w_i z_i \geq 0 \right], \quad w_i := x_i \in \mathbb{Z}.$$

Applying Lemma D.3 to $g_{\mathbf{x}}$, we obtain

$$N_{\mathbf{x}} = \binom{n}{k} 2^{n-1} \mathbb{E}_{S:\,|S|=k} \Pr\left[ g_{\mathbf{x}}(Z) \neq g_{\mathbf{x}}(Z^{\oplus S}) \right] \leq C_0 \sqrt{\frac{k}{n}} \binom{n}{k} 2^{n-1}.$$

Recalling that

$$S_1 \leq 4 \max_x N_{\mathbf{x}},$$

we conclude

$$S_1 \leq 4 \cdot C_0 \sqrt{\frac{k}{n}} \binom{n}{k} 2^{n-1} = 2C_0 \sqrt{\frac{k}{n}} \binom{n}{k} 2^n.$$

Thus there is an absolute constant $C := 2C_0 > 0$ such that

$$S_1 \leq C \sqrt{\frac{k}{n}} \binom{n}{k} 2^n = O\left( \sqrt{\frac{k}{n}} \binom{n}{k} 2^n \right).$$

$\square$

### D.2.3. PROOF OF THEOREM 6.2

Now, we give a proof of Theorem 6.2. This is established by the following theorem:

**Theorem D.5.** *Let $\mathbf{z} \in \{-1, 1\}^n$ be an unknown string and we have oracle $O_{\mathbf{z}}|\mathbf{x}, \mathbf{y}\rangle = |\mathbf{x}, \mathbf{y} \oplus f(\mathbf{x}, \mathbf{z})\rangle$, where $\mathbf{x} \in \mathbb{Z}^n$ and $f(\mathbf{x}, \mathbf{z}) = 1$ when $\langle \mathbf{x}, \mathbf{z} \rangle \geq 0$, $f(\mathbf{x}, \mathbf{z}) = 0$ when $\langle \mathbf{x}, \mathbf{z} \rangle < 0$. Then every algorithm output $\mathbf{z}$ within $k$ errors uses at least $\frac{1}{4} \log \frac{n}{k}$ queries to $O_{\mathbf{z}}$.*

*Proof.* Define $M = \{(\mathbf{x}, \tilde{\mathbf{x}}) \mid H(\mathbf{x}, \tilde{\mathbf{x}}) = k\}$ and we can get $|M| = \binom{n}{k} \cdot 2^n$. By Lemma D.4, we have

$$S_1 := \sum_{(\mathbf{x}, \tilde{\mathbf{x}}) \in M} \|O_x|\psi\rangle - O_{\tilde{\mathbf{x}}}|\psi\rangle\|^2 \leq C \sqrt{\frac{k}{n}} \binom{n}{k} 2^n.$$

We hope to get a bound of

$$S_T := \sum_{(\mathbf{x}, \tilde{\mathbf{x}}) \in M} \|O_{\mathbf{x}} U_{T-1} O_{\mathbf{x}} \cdots U_1 O_{\mathbf{x}}|\psi\rangle - O_{\tilde{\mathbf{x}}} U_{T-1} O_{\tilde{\mathbf{x}}} \cdots U_1 O_{\tilde{\mathbf{x}}}|\psi\rangle\|^2.$$

In fact, we have

$$\sum_{(\mathbf{x},\tilde{\mathbf{x}})\in M} \|O_{\mathbf{x}}U_1 O_{\mathbf{x}}|\psi\rangle - O_{\tilde{\mathbf{x}}}U_1 O_{\tilde{\mathbf{x}}}|\psi\rangle\|^2 \le 2 \sum_{(\mathbf{x},\tilde{\mathbf{x}})\in M} \|(O_{\mathbf{x}} - O_{\tilde{\mathbf{x}}})U_1 O_x|\psi\rangle\|^2 + \|O_{\mathbf{x}}|\psi\rangle - O_{\tilde{\mathbf{x}}}|\psi\rangle\|^2.$$

Assume that $|\psi\rangle = \sum_i \alpha_i |\psi_i\rangle|\phi_i\rangle$. Since our oracle can only judge, which means for different $\mathbf{x}$, $O_{\mathbf{x}}|\psi_i\rangle|\phi_i\rangle$ only have two results $|\psi_i\rangle|\phi_{i,0}\rangle$ and $|\psi_i\rangle|\phi_{i,1}\rangle$. So we can get

$$\begin{aligned}
\sum_{(\mathbf{x},\tilde{\mathbf{x}})\in M} \|(O_{\mathbf{x}} - O_{\tilde{\mathbf{x}}})U_1 O_x|\psi\rangle\|^2 &\le \sum_{(\mathbf{x},\tilde{\mathbf{x}})\in M} \sum_i \alpha_i^2 \|(O_{\mathbf{x}} - O_{\tilde{\mathbf{x}}})U_1 O_{\mathbf{x}}|\psi_i\rangle|\phi_i\rangle\|^2 \\
&\le \sum_{(\mathbf{x},\tilde{\mathbf{x}})\in M} \sum_i \alpha_i^2 (\|(O_{\mathbf{x}} - O_{\tilde{x}})U_1|\psi_i\rangle|\phi_{i,0}\rangle\|^2 + \|(O_{\mathbf{x}} - O_{\tilde{x}})U_1|\psi_i\rangle|\phi_{i,1}\rangle\|^2) \\
&= \sum_i \alpha_i^2 \sum_{(\mathbf{x},\tilde{\mathbf{x}})\in M} \|(O_{\mathbf{x}} - O_{\tilde{\mathbf{x}}})U_1|\psi_i\rangle|\phi_{i,0}\rangle\|^2 + \|(O_{\mathbf{x}} - O_{\tilde{\mathbf{x}}})U_1|\psi_i\rangle|\phi_{i,1}\rangle\|^2 \\
&\le \sum_i 2\alpha_i^2 S_1 = 2S_1.
\end{aligned}$$

Similarly, we can get

$$S_T \le (2^{T+1} + 2)S_1 \le (2^{T+1} + 2) \cdot \sqrt{\frac{k}{n}} \binom{n}{k} 2^n.$$

We let $T = \frac{1}{4}\log\frac{n}{k}$, and then there exist a $(\mathbf{x}, \tilde{\mathbf{x}}) \in M$ such that

$$\|O_{\mathbf{x}}U_{T-1}O_{\mathbf{x}} \cdots U_1 O_{\mathbf{x}}|\psi\rangle - O_{\tilde{\mathbf{x}}}U_{T-1}O_{\tilde{\mathbf{x}}} \cdots U_1 O_{\tilde{\mathbf{x}}}|\psi\rangle\|^2 \le \frac{1}{(n/k)^{\frac{1}{4}}}.$$

$\square$

Note that we can reduce Problem 1.2 to the special case of assuming $\nabla f \in \{-1, 1\}^n$, and the result allows $\varepsilon n$-bit error. By Theorem D.5, set $k = \varepsilon n$, we obtain the quantum lower bound $\Omega(\log(1/\varepsilon))$ for Theorem 6.2.

