# OpenReview forum: "Gradient Testing and Estimation by Comparisons"
_ICML.cc/2026/Conference — ICML 2026 regular_

### Official Review · Reviewer_8Hq9 · 2026-02-21

**Soundness:** 3
**Presentation:** 3
**Significance:** 2
**Originality:** 3
**Overall Recommendation:** 5
**Confidence:** 3

**Summary:**

This paper studies the question of testing/estimating the gradient of an unknown $L$-Lipschitz function $f: \mathbb{R}^n\to\mathbb{R}$ given only comparison access to it. That is, we can query whether $f(x) > f(y)$ for any two $x,y\in\mathbb{R}$. The main result are algorithms using $O(1/\epsilon)$ queries that determines whether the gradient direction of $f$ is close to or far from a particular unit vector $v$ and an algorithm that estimates the gradient of $f$ using $O(nlog(1/\epsilon))$ queries. Here we regard $L$ as a constant, so that the results are sensible and scale-invariant as they should be.

They also provide efficient quantum algorithms for the same task, where the quantum algorithm can make queries in superposition. In this review we will focus on the classical algorithms.

The testing algorithm is straightforward. Without loss of generality the test vector $v$ can be taken to be $(1,\dots,0)$. Then the tester just needs to distinguish between the gradient having large and small mass on the last $n-1$ coordinates. This can be done by rotating the last $n-1$ coordinates in a random way, using the comparison oracle to determine if that direction had enough of an effect. Applying some standard concentration inequalities proves correctness.

The gradient estimation algorithm follows by first designing an algorithm to achieve constant overlap with the actual gradient. This algorithm is simple: rotate everything by a Haar random orthogonal transformation, and learn the signs of the gradient in each coordinate direction. Then invert. The step of refining this estimate requires some more work with each coordinate.

They also prove a few lower bounds that show for example that randomness in the testing algorithm is needed to avoid an $n$ dependence in the query complexity, and that the query complexity of the gradient estimation algorithm is tight.

**Compliance With Llm Reviewing Policy:**

Affirmed.

**Final Justification:**

I will keep my positive evaluation since the authors have fully resolved concerns in the rebuttal.

**Key Questions For Authors:**

What is the motivation for not allowing direct computation of function values?

Not really a question, but it would be helpful if you could clarify the scale-invariance of the problem more explicitly.

Could you perhaps use simpler rotations than fully Haar-random rotations? These could be quite computationally expensive right?

**Limitations:**

yes

**Strengths And Weaknesses:**

Strengths:
The algorithm and analysis are fairly simple and well-motivated. The paper is well-written and addresses a good problem in optimization. The techniques are nice and clean. The algorithmic ideas could be of independent interest.

Weaknesses:
While the motivation for the problem is good, it is a bit niche and the model is a bit restrictive.

---

> ### Author Rebuttal · Authors · 2026-03-31
>
> Thank you for your positive feedback and detailed comments!
>
> **Answer to Question 1** We thank the reviewer for this question. Our motivation is not to exclude function values per se, but to study a weaker and practically relevant feedback model in which only relative preference between two points is available or reliable.
>
> Such comparison-only feedback arises naturally in preference-based optimization, dueling/A-B style evaluation, and other settings where ordinal judgments are easier to obtain than well-calibrated scalar values. Moreover, even when a numerical score exists, it may be noisy or poorly calibrated, while pairwise ordering is often more robust.
>
> From a theoretical perspective, comparison-based optimization is a natural model that has been studied in the optimization literature, see, e.g., [1,2].
>
> [1] El Houcine Bergou, Eduard Gorbunov, Peter Richtárik. Stochastic Three Points Method for Unconstrained Smooth Minimization.
>
> [2] Eduard Gorbunov, Adel Bibi, Ozan Sener, El Houcine Bergou, and Peter Richtárik. A stochastic derivative free optimization method with momentum.
>
> **Answer to Question 2** We thank the reviewer for pointing this out.
> In the comparison oracle model, rescaling of the objective does not affect pairwise comparison outcomes. Thus, both the oracle access pattern and the target quantity are invariant to the overall scale of the function values.
>
> We will make this intuition more explicit in the revised version. If the reviewer had another notion of scale-invariance in mind, we would greatly appreciate further clarification, and we would be happy to address it carefully.
>
> **Answer to Question 3** We thank the reviewer for this insightful suggestion. We agree that fully Haar-random rotations may be unnecessarily strong from an implementation perspective.
>
> In fact, our algorithms do not require explicitly generating or applying a full Haar-random orthogonal matrix. What is used in the analysis is a random direction, together with a coordinate change that maps the testing direction to $e_1$. The latter is simply a convenient reparameterization of the analysis, and can be implemented by any orthogonal transformation sending the chosen direction to $e_1$; it is not essential that this transformation itself be Haar-random.
>
> More generally, we believe the proof only uses anti-concentration properties of the random direction, rather than the full strength of Haar randomness. This suggests that weaker random objects, for example, suitable approximate unitary 2-designs, may already suffice. We agree it is a very interesting future direction, and will add more discussions on this in the final version.

---

> > ### Author Rebuttal · Reviewer_8Hq9 · 2026-03-31
> >
> > I thank the authors for their thoughtful rebuttal.

---

> > > ### Author Response · Authors · 2026-04-01
> > >
> > > Thank you very much for your very positive feedback and for raising your recommendation to 5. We are delighted that our response fully resolved your concerns.

---

### Official Review · Reviewer_Kp2P · 2026-02-27

**Soundness:** 2
**Presentation:** 2
**Significance:** 2
**Originality:** 3
**Overall Recommendation:** 5
**Confidence:** 4

**Summary:**

The problem explored by the study is how to efficiently perform gradient testing and gradient estimation for smooth functions when restricted to a minimal zeroth-order comparison oracle that only reveals which of two queried points has a larger function value. The central technical challenge of the paper lies in establishing the fundamental query complexity limits for extracting this precise geometric information in both classical and quantum computing paradigms. To this end, the authors propose classical algorithms that achieve $O(n)$ query complexity for gradient testing and $O(n \log(1/\varepsilon))$ for gradient estimation, alongside lower bound proofs asserting the strict optimality of both algorithms. Furthermore, by allowing the comparison oracle to be queried in superposition, the paper extends the study to the quantum setting, developing quantum algorithms that reduce the query complexities to $O(1)$ for gradient testing and $O(\log(n/\varepsilon))$ for gradient estimation.

**Compliance With Llm Reviewing Policy:**

Affirmed.

**Final Justification:**

Although the initial version of the paper had several issues, the authors have made substantial and thoughtful revisions in response to the feedback. The rebuttal effectively addressed my main concerns, significantly improving the soundness and clarity of the work. I am now satisfied with the current version and believe it represents a strong and valuable contribution.

**Key Questions For Authors:**

Please refer to the 'Strengths and Weaknesses' for my specific questions and technical concerns.

**Limitations:**

yes

**Strengths And Weaknesses:**

The paper studies the query complexity of gradient testing and estimation under a comparison oracle. On the positive side, the authors provide a relatively complete theoretical picture of this specific problem setting. They analyze both testing and estimation tasks across classical and quantum regimes, and provide corresponding upper and lower bounds. This comprehensive treatment of the theoretical problem is the main strength of the paper.

However, the paper lacks a clear motivation for the practical significance of its problem setting. The authors focus strictly on estimating or testing the gradient, but fail to justify studying these primitives in isolation from a broader optimization trajectory. In zeroth-order settings, the ultimate goal is to optimize the objective function, and estimating or testing the gradient is merely a means to that end. It is entirely unclear how the proposed algorithms perform within an actual optimization loop. In contrast, prior related works cited by the authors, such as Karabag et al. (2021), Cai et al. (2022), and Tang et al. (2024), explicitly focus on the overall optimization dynamics and substantiate their theoretical claims with practical applications and empirical experiments, which are notably absent here.

Furthermore, there is a critical flaw in the technical soundness of the classical lower bound for gradient testing (Theorem 5.1). The claim that any classical algorithm requires $\Omega(n)$ queries overlooks the power of simple randomized algorithms. Consider a randomized algorithm that queries a test direction $(-k, u)$, where $k \approx \epsilon/\sqrt{n}$ serves as a negative threshold along the target axis, and $u$ is a random unit vector uniformly sampled from the $(n-1)$-dimensional orthogonal subspace. For a "YES" instance ($g=e_1$), the oracle response is deterministically negative. For a "NO" instance where the gradient is tilted toward some $e_i$, the variance of $u_i$ ensures that the oracle response flips with a constant probability gap (roughly $p \approx 0.69$ vs $p=1$). Consequently, by the Chernoff bound, the testing problem can be solved with $O(1)$ classical queries. This directly invalidates the claimed $\Omega(n)$ lower bound and demonstrates that an $O(1)$ classical algorithm already exists.

As a direct consequence of the counter-example above, it appears that the quantum algorithm for this specific problem can be dequantized, thereby yielding an equivalent $O(1)$ classical algorithm. The loophole in the authors' classical lower bound proof lies precisely in the proof of Lemma C.1 (around line 1040). The proof erroneously deduces that $|S| \le O(1)$ from the inequality $|S| \cdot c^2|w_1|^2 \le 1$. This deduction is mathematically invalid because it implicitly assumes that the query vector's first coordinate $|w_1|$ is a constant strictly bounded away from zero. However, an algorithm can choose a "dense" query vector that intentionally shrinks $w_1$ to the scale of $1/\sqrt{n}$ while distributing the remaining $L_2$ mass evenly across all orthogonal dimensions. With Lemma C.1 invalidated, the subsequent union bound and the application of Yao's Minimax Principle collapse entirely.

Beyond the theoretical aspects above, the presentation of the paper could also be significantly improved. The main text currently contains many detailed coefficient calculations but lacks sufficient explanation of the algorithmic ideas, making it somewhat difficult for a broader audience to follow. I would suggest relegating these dense mathematical proof details to the appendix, and instead, adding more high-level explanations and intuitions regarding the algorithm design in the main text.

---

> ### Author Rebuttal · Authors · 2026-03-31
>
> Thank you for your very detailed feedback and comments!
>
>  **Response to Weakness 1** We thank the reviewer for this suggestion. We conducted numerical experiments; given that the rebuttal has 5000-character limit, below we very briefly summarize our experimental results.
>
> On one hand, to verify correctness, we evaluated our gradient testing and estimation algorithm on various functions (e.g., Extended Rosenbrock, Quadratic). Remarkably, for both gradient testing (including our initial $O(n)$ algorithm and revised $O(1)$ randomized algorithm) and estimation, our algorithms achieve a near 100\% empirical success rate, far exceeding our theoretical worst-case guarantees. For gradient estimation, the empirical query complexity perfectly aligns with our theoretical $O(n \log(1/\epsilon))$ bound.
>
> On the other hand, to demonstrate practical utility, we integrated our gradient estimator into the adaptive normalized gradient descent (AdaNGD) framework. We compared it against the Ideal AdaNGD (Levy, 2017) with exact gradient direction, ZO-SGD (Nesterov \& Spokoiny, 2017), ZO-RankSGD (Tang et al., 2024), and SCOBO (Cai et al., 2022), under a matched query budget per iteration. Across different settings (with and without line search), AdaNGD equipped with our estimator achieves a convergence remarkably close to the Ideal AdaNGD and significantly outperforms other baselines. For example, on the Extended Rosenbrock with line search, our method reaches an objective value of 0.55 at iteration 190 (closely tracking Ideal NGD's 0.54), whereas ZO-SGD, ZO-RankSGD, and SCOBO only reach 5.38, 59.94, and 69.46, respectively. This demonstrates the effectiveness of our high precision gradient estimation in optimization tasks.
>
> We are happy to present more details of our numerical experiments after the start of the reviewer-author discussion period.
>
>  **Response to Weakness 2** We very much thank the reviewer for spotting flaw. We agree that the statement in the submitted version of Theorem 5.1 is imprecise. In fact, after the submission deadline and before the current rebuttal starts, we identified this issue.
> The corrected statement is that gradient testing admits an *$O(1)$-query classical randomized algorithm*, while the *deterministic classical query complexity is $\Theta(n)$*. Given that the rebuttal has 5000-character limit, below we very briefly summarize our intuition.
>
> On the one hand, for the **randomized** case, there is an $O(1)$-query algorithm: in each round, sample a random unit vector $y$ from the orthogonal subspace of $v$, construct a probe direction
> $$
> \alpha_y = \Bigl(-\frac{\epsilon}{\sqrt{(n-1)(1-\epsilon^2)}}, y\Bigr),
> $$
> and use the directional-preference subroutine to record the returned sign. The procedure tests whether the gradient is sufficiently aligned with $v$ or has a noticeable orthogonal component. In the revised analysis, the probability of a negative outcome has a constant gap between the YES and NO cases, so a **constant** number of repetitions suffices. In this sense, the reviewer’s probing idea is very much aligned with the corrected randomized algorithm.
>
> On the other hand, for the **deterministic** case, our $\Omega(n)$ lower bound remains valid. The intuition of the lower bound is that, for any deterministic algorithm making fewer than $n-1$ queries, the queried directions do not determine the unknown gradient uniquely: there still exist both a YES instance and a NO instance that are consistent with exactly the same transcript on all the queries.
>
> In addition, we also revise Algorithm 2 to give an $O(n)$ deterministic upper bound. The main change is that the original random step for certifying that the anchor coordinate is not too small can be replaced by a fixed query direction $y=(2n,-1,\dots,-1)$. If the corresponding DP query shows that this anchor is not sufficiently dominant, the algorithm can deterministically conclude that the input is in the NO case. Together with the lower bound, this yields the tight deterministic complexity $\Theta(n)$.
>
> So the accurate conclusion is:
>
> **randomized** classical testing $=O(1)$,  **deterministic** classical testing $=\Theta(n)$.
> ---
>
> All these revisions will be merged in the final version of our paper. We are happy to present more details after the start of the reviewer-author discussion period (when there's no character limitation).
>
>  **Response to Weakness 3** We thank the reviewer for this helpful suggestion. In the revision, we will make the main text more focused on the algorithmic insights and intuition behind our results. Specifically, we will shift more proof-level coefficient calculations and other routine derivations to the appendix, and correspondingly expand the high-level explanation of the algorithm design in the main body. For both the classical and quantum parts, we will add more intuition on the main primitives, clarify the role played by each step of the procedure, and include short roadmaps before the technical arguments.

---

> > ### Author Rebuttal · Reviewer_Kp2P · 2026-04-01
> >
> > Thank you for the careful and thoughtful response. I believe it fully resolves my concerns. I am therefore happy to raise my recommendation to 5, and I look forward to seeing a revised and improved final version of the paper.

---

> > > ### Author Response · Authors · 2026-04-01
> > >
> > > Thank you so much for your very positive feedback and for raising your recommendation to 5. We are delighted that our response fully resolved your concerns.
> > >
> > > For your reference, we have attached a brief experimental note below that provides additional details related to the comparison in optimization tasks.
> > >
> > > Specifically, we apply our gradient estimation at each iteration of AdaNGD, which is originally introduced in [3]. We compare the convergence performance of AdaNGD with our gradient estimation against the ideal AdaNGD, ZO-SGD introduced in [4], ZO-RankSGD introduced in [5] and SCOBO introduced in [1]. ZO-SGD use multiple zeroth-order queries to obtain a gradient approximation in each iteration, while both ZO-RankSGD and SCOBO use multiple comparison queries in each iteration to obtain a gradient approximation, but the accuracy is relatively low. The work by [2] also utilizes comparison queries for optimization, but their complexity in terms of $n$ is significantly higher, and thus we did not include it in our comparison.
> > >
> > > In our experiments, we assume that all these methods use a comparable number of queries of comparison (or function value) in each iteration. Specifically, we consider the Extended Rosenbrock with dimension $n=100$: $f(x) = \sum_{i=1}^{99} [100(x_{i+1} - x_i^2)^2 + (x_i - 1)^2]$. We assume that in each iteration, ZO-SGD queries the function value at 100 random perturbations to obtain a gradient approximation; ZO-RankSGD obtains the full ranking of function values among 100 points; meanwhile, SCOBO receives the comparison results between the current point $x$ and 100 perturbed neighbors.
> > >
> > > We first conduct experiments without line search. The step size is set to $\eta_t=R/\sqrt{2t}$ for AdaNGD with $R=5$, $\eta = 0.01$ for ZO-SGD (since the function value is large), and $\eta=0.5$ for both ZO-RankSGD and SCOBO. The experimental results are shown in Table 4. We can see that AdaNGD with our gradient estimation achieves a convergence performance that is very close to the ideal AdaNGD, and significantly outperforms ZO-SGD, ZO-RankSGD and SCOBO. This demonstrates the effectiveness of our high precision gradient estimation in optimization tasks.
> > >
> > > We also conduct experiments with line search, which is commonly used in practice and also adopted in [1] and [5]. The results are shown in Table 5. We can see that our method still significantly outperforms the baselines.
> > >
> > > **Reference**
> > >
> > > [1] Cai, H., McKenzie, D., Yin, W., and Zhang, Z. A one-bit, comparison-based gradient estimator.
> > >
> > > [2] Karabag, M. O., Neary, C., and Topcu, U. Smooth convex optimization using sub-zeroth-order oracles.
> > >
> > > [3] Levy, K. Online to offline conversions, universality and adaptive minibatch sizes.
> > >
> > > [4] Nesterov, Y. and Spokoiny, V. Random gradient-free minimization of convex functions.
> > >
> > > [5] Tang, Z., Rybin, D., and Chang, T.-H. Zeroth-order optimization meets human feedback: Provable learning via ranking oracles.
> > >
> > > ---
> > >
> > > **Table 4**: *Comparison of AdaNGD with Our Gradient Estimation against Ideal AdaNGD, ZO-SGD, ZO-RankSGD and SCOBO. We consider the Extended Rosenbrock function with $d=100$. The step size is set to $\eta_t=R/\sqrt{2t}$ for AdaNGD with $R=5$, $\eta = 0.01$ for ZO-SGD, and $\eta=0.5$ for ZO-RankSGD and SCOBO.*
> > >
> > > | Iteration | Ideal AdaNGD [3]  | ZO-SGD [4]  | Ours: AdaSGD with Gradient Estimation  | ZO-RankSGD [5]  | SCOBO [1]  |
> > > | :--- | :--- | :--- | :--- | :--- | :--- |
> > > | 0 | 108259.24 | 108259.24 | 108259.24 | 108259.24 | 108259.24 |
> > > | 20 | 65.12 | 4071.53 | 61.18 | 1082.83 | 12144.47 |
> > > | 40 | 18.46 | 1202.02 | 16.48 | 204.99 | 267.30 |
> > > | 60 | 7.16 | 491.64 | 6.13 | 253.86 | 104.50 |
> > > | 80 | 3.16 | 232.89 | 8.79 | 209.58 | 99.01 |
> > > | 100 | 6.10 | 125.22 | 5.67 | 206.39 | 92.23 |
> > > | 120 | 4.45 | 70.29 | 3.79 | 254.35 | 101.21 |
> > > | 140 | 3.10 | 43.18 | 2.59 | 189.48 | 83.58 |
> > > | 160 | 2.18 | 27.09 | 1.82 | 267.02 | 95.62 |
> > > | 180 | 1.54 | 18.72 | 1.34 | 251.71 | 100.80 |
> > >
> > > ---
> > >
> > > **Table 5**: *Comparison of NGD with Our Gradient Estimation against Ideal NGD, ZO-SGD, ZO-RankSGD and SCOBO with line search. We consider the Extended Rosenbrock function with $d=100$. The step size is set to $\eta = 0.01$ for ZO-SGD and $\eta=50$ for others. In each iteration the line search is performed 5 times. We set the shrinking rate $\gamma = 0.1$ and number of trails to 5 for line search.*
> > >
> > > | Iteration | Ideal AdaNGD [3]  | ZO-SGD [4]  | Ours: AdaSGD with Gradient Estimation  | ZO-RankSGD [5]  | SCOBO [1]  |
> > > | :--- | :--- | :--- | :--- | :--- | :--- |
> > > | 0 | 106116.71 | 106116.71 | 106116.71 | 106116.71 | 106116.71 |
> > > | 20 | 1.40 | 227.40 | 1.36 | 454.61 | 828.60 |
> > > | 40 | 0.57 | 47.76 | 0.57 | 168.41 | 262.25 |
> > > | 60 | 0.56 | 28.13 | 0.57 | 99.96 | 196.34 |
> > > | 80 | 0.56 | 21.03 | 0.56 | 69.15 | 148.51 |
> > > | 100 | 0.56 | 14.85 | 0.56 | 65.73 | 113.60 |
> > > | 120 | 0.55 | 10.78 | 0.56 | 64.42 | 84.75 |
> > > | 140 | 0.55 | 8.57 | 0.55 | 63.27 | 71.53 |
> > > | 160 | 0.55 | 6.49 | 0.55 | 62.07 | 69.72 |
> > > | 180 | 0.55 | 5.73 | 0.55 | 60.69 | 69.49 |

---

### Official Review · Reviewer_wgQG · 2026-03-02

**Soundness:** 3
**Presentation:** 3
**Significance:** 3
**Originality:** 3
**Overall Recommendation:** 5
**Confidence:** 3

**Summary:**

This paper studies gradient testing and gradient estimation when only comparison oracle access is available, so the algorithm can only compare function values at two points. The main challenge addressed by this paper is determining how much gradient information can be recovered under such minimal feedback.

The authors propose optimal classical algorithms for gradient testing and estimation together with matching lower bounds, and develop quantum algorithms achieving improved query complexity without achieving the matching lower bound for gradient estimation.

**Compliance With Llm Reviewing Policy:**

Affirmed.

**Ethical Review Concerns:**

No concerns

**Key Questions For Authors:**

Here are a few questions:

How sensitive are the guarantees when gradients become small near stationary points?

Can the proposed procedures be incorporated into a full optimization algorithm with convergence guarantees?

Is the remaining logarithmic gap in quantum gradient estimation fundamental, or could it be removable?

Can the authors provide some (experimental) evidence that they algorithms work well in practice?

**Limitations:**

The contributions are primarily theoretical and do not involve direct real-world applications.

**Strengths And Weaknesses:**

Strength:

 The work is technically solid. Algorithms are carefully designed and supported by rigorous analysis, and matching lower bounds establish optimality. Assumptions are clearly stated and proofs appear sound.

The paper is well organized and generally clear. The introduction motivates the comparison-oracle setting effectively, and the structure separating intuition from formal analysis improves readability.

The paper addresses a meaningful theoretical question in derivative-free optimization and establishes tight complexity results.

The formulation of gradient testing in the comparison-oracle model and the unified classical and quantum analysis provide clear novelty compared to previous related literature.

Weakness:

The guarantees rely on a lower bound assumption $\|\nabla f(x)\|\ge \gamma$, which restricts applicability near stationary points and makes the connection to practical optimization somewhat indirect.

Additionally, the quantum gradient estimation result is optimal only up to a logarithmic factor, leaving a gap between the upper and lower bounds whose necessity remains unclear.

 In case of the quantum estimations there are several novel algorithms, including by Heidari et al. "Hadamard Test is Sufficient for Efficient Quantum Gradient Estimation with Lie Algebraic Symmetries" published in 2025 NeurIPS, that may use for a comparison even in the underlying settings are not the same.

There are no experimental results so it hard to judge how practical the proposal algorithms are. This is a serious drawback.

---

> ### Author Rebuttal · Authors · 2026-03-31
>
> Thank you for your positive feedback and detailed comments!
>
> **Answer to Question 1** Thank you for raising this point. We agree that near stationary points, directional guarantees should be interpreted with care.
>
> Our guarantees are stated under the promise $\|\nabla f(x)\|\ge \gamma$, since in the comparison-only model one can recover only gradient *direction*, not magnitude; When $\|\nabla f(x)\|$ becomes too small near a stationary point, the directional signal becomes indistinguishable from the smoothness error at the comparison scale.
>
> Importantly, however, our query complexity does not deteriorate as $\gamma$ becomes small: $\gamma$ is used only to set the precision parameters in the directional-preference subroutine, while the overall asymptotic complexities remain unchanged. Thus the guarantees remain valid for arbitrarily small $\gamma>0$, as long as the promise $\|\nabla f(x)\|\ge\gamma$ holds. In this sense, the results are not especially sensitive to small gradients at the level of query complexity; rather, $\gamma$ specifies the regime in which recovering a descent-relevant direction is meaningful. We will clarify this point in the revision.
>
> **Answer to Question 2** We thank the reviewer for this question. Yes, our procedures can be incorporated into a full optimization algorithm by using the estimated direction as a plug-in replacement for the normalized gradient in normalized gradient descent (NGD). If at iterate $x_t$ our estimator outputs a unit vector $\hat g_t$ with $\|\hat g_t-g_t\|\le \varepsilon$, then
> $$
> \langle g_t,\hat g_t\rangle \ge 1-\frac{\varepsilon^2}{2},
> $$
> so $\hat g_t$ remains positively aligned with the true normalized gradient. Therefore, the update
> $$
> x_{t+1}=x_t-\eta_t \hat g_t
> $$
> is an inexact NGD step, and by $L$-smoothness,
> $$
> f(x_{t+1})\le f(x_t)-\eta_t\|\nabla f(x_t)\|\Bigl(1-\frac{\varepsilon^2}{2}\Bigr)+\frac{L}{2}\eta_t^2.
> $$
> Thus, whenever $\|\nabla f(x_t)\|\ge \gamma$ and $\eta_t$ is chosen appropriately, each step yields guaranteed descent. We will make this connection explicit in the revision and discuss a full end-to-end convergence analysis as a natural future direction; see also [1].
>
> [1] Levy, K. Online to offline conversions, universality and adaptive minibatch sizes. Advances in Neural Information Processing Systems, 30, 2017. arXiv:1705.10499
>
> **Answer to Question 3** We thank the reviewer for this insightful question. At present, we do not know whether the remaining logarithmic gap is fundamental. Our lower bound is $\Omega(\log(1/\varepsilon))$, while our algorithm achieves $O(\log(n/\varepsilon))$, leaving only a $\log n$ gap. We believe this factor may be removable, but likely not within the current binary-search-plus-QFT framework.
>
> The extra $\log n$ term comes from identifying an orthogonality/scale parameter with sufficient precision before the final QFT decoding step. We explored using geometric concentration on the unit sphere to reduce this search range. However, this is not enough to maintain the precise phase structure required for full $\varepsilon$-accurate QFT-based reconstruction. We will clarify this point and highlight removing the $\log n$ factor as an interesting open problem.
>
> **Answer to Question 4** We thank the reviewer for this suggestion. We conducted numerical experiments; given that the rebuttal has 5000-character limit, below we very briefly summarize our experimental results.
>
> On one hand, to verify correctness, we evaluated our gradient testing and estimation algorithm on various functions (Extended Rosenbrock, Quadratic). Remarkably, for both testing (including our $O(n)$ algorithm and revised $O(1)$ randomized algorithm) and estimation, our algorithms achieve a near 100\% empirical success rate, far exceeding our theoretical worst-case guarantees. For gradient estimation, the empirical query complexity perfectly aligns with our theoretical $O(n \log(1/\epsilon))$ bound.
>
> On the other hand, to demonstrate practical utility, we integrated our gradient estimator into the adaptive normalized gradient descent (AdaNGD) framework. We compared it against the Ideal AdaNGD (Levy, 2017) with exact gradient direction, ZO-SGD (Nesterov \& Spokoiny, 2017), ZO-RankSGD (Tang et al., 2024), and SCOBO (Cai et al., 2022), under a matched query budget per iteration. Across different settings (with and without line search), AdaNGD with our estimator achieves a convergence remarkably close to the Ideal AdaNGD and significantly outperforms other baselines. For example, on Extended Rosenbrock with line search, our method reaches an objective value of 0.55 at iteration 190 (closely tracking Ideal NGD's 0.54), whereas ZO-SGD, ZO-RankSGD, and SCOBO only reach 5.38, 59.94, and 69.46, respectively. This demonstrates the effectiveness of our high precision gradient estimation in optimization tasks.
>
> We are happy to present more details of our numerical experiments after the start of the reviewer-author discussion period.

---

> > ### Author Rebuttal · Reviewer_wgQG · 2026-04-01
> >
> > Thanks for a detailed response. I raise the score to "accept".

---

> > > ### Author Response · Authors · 2026-04-01
> > >
> > > Thank you very much for your very positive feedback and for raising your recommendation to 5. We are delighted that our response fully resolved your concerns.
> > >
> > > For your reference, additional details about our experiments are provided in our response to Reviewer Kp2P.

---

### Official Review · Reviewer_UENq · 2026-03-11

**Soundness:** 3
**Presentation:** 3
**Significance:** 3
**Originality:** 3
**Overall Recommendation:** 4
**Confidence:** 4

**Summary:**

This paper solves two fundamental problems in testing and estimating gradient by comparison oracles, which is of great importance in nowdays LLMs training. The paper provides optimal classical and quantum methods for these two tasks.

**Compliance With Llm Reviewing Policy:**

Affirmed.

**Final Justification:**

The authors have addressed my concern and I decide to remain my orginal score.

**Key Questions For Authors:**

1. The estimation of gradient has widely applications. I suggest  the authors to add some experiments to show how well their constructed gradient estimator approximates the true gradient, and the comparison with zeroth-order estimation to the gradient.

2. The authors solve the problems with probability at least 2/3. I am wondering what is the complexity if we want a high successful probablity of $1-\delta$? I suggest the authors to add results for this case, so that these estimators can be immediately applied to optimization tasks.

**Limitations:**

Please refer to the question part.

**Strengths And Weaknesses:**

# Strengths
1. The problem is fundamental and important to solve. Testing gradient and estimating gradient with only comparison oracles enable us to train the model by the gradient with only simple comparison feedback.
2. This paper provide useful tools to test and estimate gradient with $\tilde{O}(n)$ classical comparison oracles or $\tilde{O}(1)$ quantum classical comparison oracles. Moreover, the paper provides lower bound for both tasks, which indicates that their methods are (near) optimal. The theoretical results in this paper is novel and strong to me.

# Weakness
Please refer to the question part.

---

> ### Author Rebuttal · Authors · 2026-03-31
>
> Thank you for your positive feedback and detailed comments!
>
> **Answer to Question 1** We thank the reviewer for this suggestion. We conducted numerical experiments; given that the rebuttal has 5000-character limit, below we very briefly summarize our experimental results.
>
> On one hand, to verify correctness, we evaluated our gradient testing and estimation algorithm on various functions (e.g., Extended Rosenbrock, Quadratic). Remarkably, for both gradient testing (including our initial $O(n)$ algorithm and revised $O(1)$ randomized algorithm) and estimation, our algorithms achieve a near 100\% empirical success rate, far exceeding our theoretical worst-case guarantees. For gradient estimation, the empirical query complexity perfectly aligns with our theoretical $O(n \log(1/\epsilon))$ bound.
>
> On the other hand, to demonstrate practical utility, we integrated our gradient estimator into the adaptive normalized gradient descent (AdaNGD) framework. We compared it against the Ideal AdaNGD (Levy, 2017) with exact gradient direction, ZO-SGD (Nesterov \& Spokoiny, 2017), ZO-RankSGD (Tang et al., 2024), and SCOBO (Cai et al., 2022), under a matched query budget per iteration. Across different settings (with and without line search), AdaNGD equipped with our estimator achieves a convergence remarkably close to the Ideal AdaNGD and significantly outperforms other baselines. For example, on the Extended Rosenbrock with line search, our method reaches an objective value of 0.55 at iteration 190 (closely tracking Ideal NGD's 0.54), whereas ZO-SGD, ZO-RankSGD, and SCOBO only reach 5.38, 59.94, and 69.46, respectively. This demonstrates the effectiveness of our high precision gradient estimation in optimization tasks.
>
> We are happy to present more details of our numerical experiments after the start of the reviewer-author discussion period.
>
> **Answer to Question 2** We thank the reviewer for raising this important point. We agree that a high-probability guarantee is particularly relevant if one wants to plug the proposed testing/estimation primitives into iterative optimization procedures.
>
> For gradient testing, the constant-success guarantees can be boosted to $1-\delta$ by majority vote. Similarly, for gradient estimation, the constant-success guarantees can be boosted to $1-\delta$ by standard repetition. One can run the estimator independently $O(\log(1/\delta))$ times and apply a standard robust aggregation step (e.g., selecting an output that has the largest number of neighboring estimates within the target accuracy scale). Since each run is $\varepsilon$-accurate with constant probability (Theorem 3.4), a Chernoff bound implies that with probability at least $1-\delta$, a strict majority of the returned vectors are good; the aggregation step then returns an $O(\varepsilon)$-accurate estimate. Equivalently, by running the base estimator with accuracy parameter $\varepsilon/c$ for a suitable constant $c$, one obtains a final $\varepsilon$-accurate estimate with success probability at least $1-\delta$, using
> $$
> O\left(n\log\frac{1}{\varepsilon}\log\frac{1}{\delta}\right)
> $$
> comparison queries.
>
> We will add a short remark/corollary discussing these high-probability extensions.

---

> > ### Author Rebuttal · Reviewer_UENq · 2026-04-01
> >
> > Thank you for answering my questions.

---

> > > ### Author Response · Authors · 2026-04-01
> > >
> > > Thank you again for your positive feedback and support.
> > >
> > > For your reference, additional details about our experiments are provided in our response to Reviewer Kp2P.

---

### Decision · Program_Chairs · 2026-04-30

**Decision:**

Accept (regular)

**Comment:**

Consider the problems of gradient testing and estimation with only a comparison oracle. This paper proposes classical and quantum algorithms with oracle complexity guarantees, as well as oracle complexity lower bounds implying the optimality or near-optimality of the proposed algorithms.

- Reviewers 8Hq9 and Kp2P raised concerns about the motivation for studying this problem setup, while Reviewers wgQG and UENq were concerned about the lack of experiments. These concerns have been addressed by the addition of numerical experiments in which the proposed gradient estimation algorithm is integrated into AdaNGD and compared with existing algorithms using comparison or zeroth-order oracles.
- Reviewer Kp2P provided a counterexample to the classical lower bound for gradient testing. The authors have revised their theorem statements accordingly.

The reviewers all agree that the paper is technically solid, after the revision of the classical lower bound for gradient testing. I believe that the results should be relevant to the optimization community. With the newly added numerical results, I am happy to recommend acceptance of this paper.